# Molecular anatomy of PLK1 master docking motifs

Long Ren ®[1], Arianna Esposito-Verza ®[1], Raphael Gasper ®[2],
Marion E. Pesenti ®[1], Petra Janning[3], Franziska Müller ®[1], Carolin Koerner[1],
Petra Geue[2], Sabine Wohlgemuth[1], Ingrid R. Vetter[1] & Andrea Musacchio ®[1,4] ✉

The Polo-box domain (PBD) localizes Polo-like kinase 1 (PLK1) near mitotic substrates required for chromosome biorientation. Recent work on mitotic kinetochores showed PLK1 docking begins hierarchically at master docking motifs on BUB1 and CENP-U. Whether master docking motifs have common molecular features remains poorly understood. Presence on CENP-U of two neighbouring motifs generated by initial CDK1 priming and subsequent PLK1 phosphorylation led us to hypothesize PBD dimerization might be involved. Using biochemical, biophysical, and modelling approaches, we gathered strong evidence that CENP-U contains a single master docking motif. The motif is very high affinity and sufficient to form extensive interactions with the PBD, engaging multiple pockets on its surface without obvious added benefits from dimerization. Comparisons with motifs in BUB1, BUBR1, and PRC1 suggest commonalities of master PLK1 docking motifs. We discuss the implications of our observations for the mechanism of PLK1 activation.

The cell cycle, the universal process of division of a mother cell into two daughters, is the foundation of cellular life. In eukaryotes, this process is executed through master regulators named cyclin-dependent kinases (CDKs), whose progressive activation orders all crucial cell cycle events, from the replication of chromosomes in S-phase to their segregation from the mother cell into its two daughters during M-phase (mitosis). CDK activation ultimately also controls the activity of additional cell cycle regulators, among which are additional mitotic kinases[1,2]. A crucial mitotic kinase, Polo-like kinase 1 (PLK1), has emerged for its essential functions in a number of cell division events[3–7]. For instance, PLK1 has been implicated in the regulation of spindle assembly, centrosome function, nuclear envelope breakdown, sister chromatid cohesion, kinetochore-microtubule interactions, spindle assembly checkpoint signalling, centromere propagation, and cytokinesis, among others[3–7]. How PLK1 controls these processes with exquisite spatial and temporal accuracy remains incompletely understood.

Human PLK1 is a 603-residue protein consisting of an amino-terminal Ser/Thr catalytic domain (KD) separated through a 67-residue interdomain linker (IDL) and the Polo Cap (PC) helix from a carboxy-terminal polo-box domain (PBD) (Fig. 1A)[8–10]. The PBD consists of two polo box (PB) subdomains, PB1 and PB2 (residues 411-489 and 511-592, respectively), each with a preceding loop, named L1 and L2 respectively (Fig. 1A)[11,12]. Each PB consists of a six-stranded anti-parallel β-sheet followed by an α-helix (β6α fold). The two PBs in a PBD form a tightly packed functional unit stabilized by multiple interactions in a shared hydrophobic core[13].

The interplay between the PLK1 KD and the PBD has attracted considerable interest and remains only partly understood. The PBD has at least two functions. First, it mediates the subcellular localization of PLK1 by binding to specific phosphorylated docking motifs in target proteins[11,12,14–16]. There is a quite strict requirement for targets to contain the sequence Ser-pThr-X or Ser-pSer-X (abbreviated respectively as S-pT-X or S-pS-X, where p indicates phosphorylation), with pThr

[1]Department of Mechanistic Cell Biology, Max Planck Institute of Molecular Physiology, Dortmund, Germany. [2]Crystallography and Biophysics Facility, Max Planck Institute of Molecular Physiology, Dortmund, Germany. [3]Mass Spectrometry Core Facility, Max Planck Institute of Molecular Physiology, Dortmund, Germany. [4]Centre for Medical Biotechnology, Faculty of Biology, University Duisburg-Essen, Essen, Germany. ✉e-mail: andrea.musacchio@mpi-dortmund.mpg.de

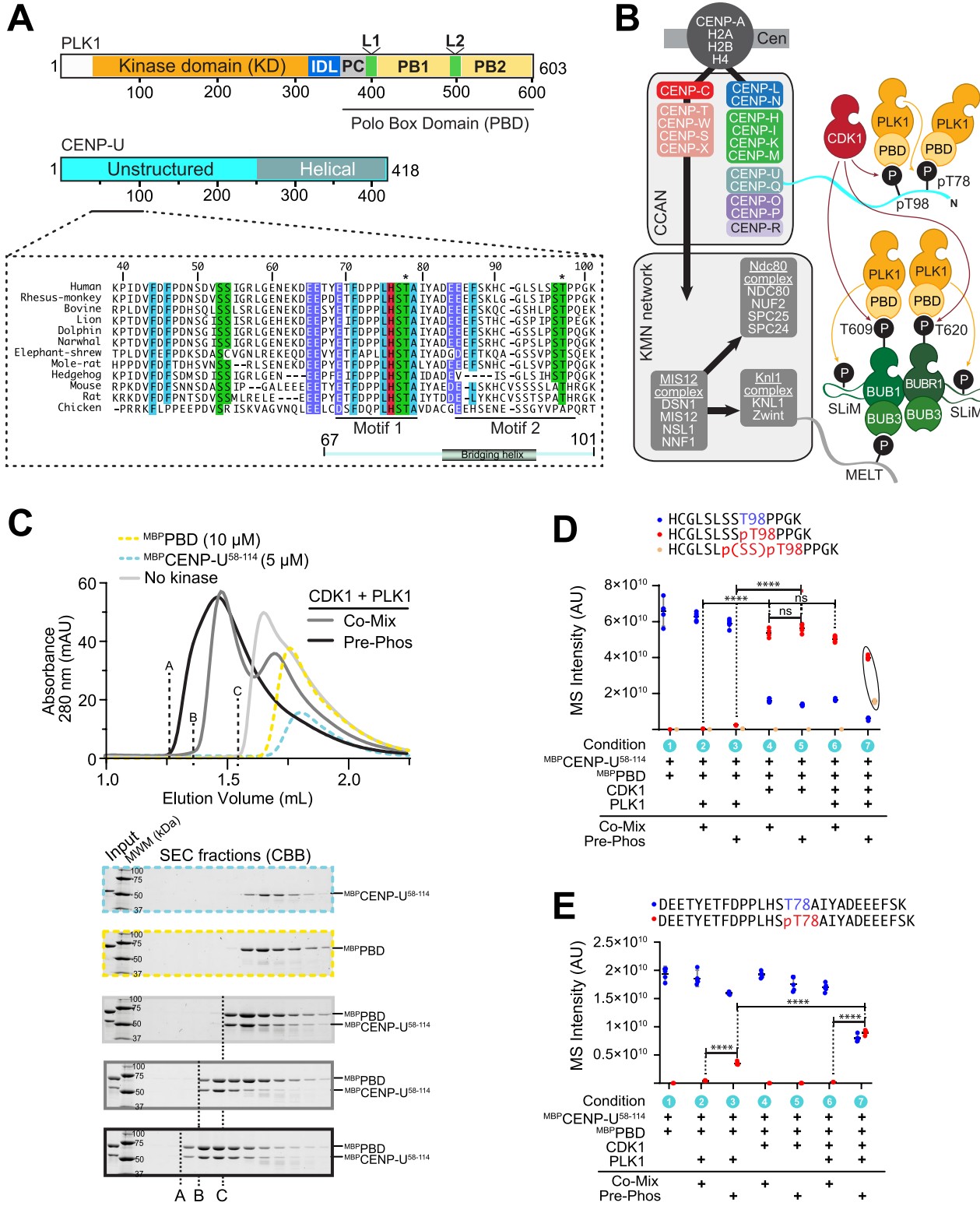

preferred over pSer[17]. The X residue following pS/pT influences PBD binding selectivity only modestly, but influences greatly the choice of kinase introducing the phosphate. When X is Pro, the motif is a target of proline-directed kinases, like the CDKs. This widely documented mechanism, named CDK priming (an instantiation of nonself priming), links mitotic CDK activation to PLK1 localization and further PLK1 target phosphorylation[17,18]. When X is not Pro, other kinases may be involved in the generation of PBD docking sites. Self-priming may occur when the phosphorylating kinase is PLK1 itself[19–22]. Reflecting the

preferences of the PLK1 kinase domain, this may occur especially in the presence of Leu at position −3 relative to the target S/T; Asp, Asn, Glu, or Gln at position −2; and in the absence of Pro at the +1 position[23–25].

Second, the PBD controls the activity of the KD through direct intramolecular interactions believed to dampen kinase activity[8,26–28]. Extensive interactions between the PBD and the KD inhibit PLK1 kinase activity in the absence of phosphorylated ligands[27–29]. The interface between the L1 loop and the kinase small lobe, and a sandwich of the interdomain-linker (IDL) between the kinase hinge domain and the L2

**Fig. 1 | Assembly of a PLK1:CENP-U complex. A** Schematic representation of the PLK1 and CENP-U. KD kinase domain, IDL inter-domain linker, PC Polo Cap, L1, L2 Loop 1 and 2, PB1 and PB2 Polo-box 1 and 2. **B** Hierarchical organization of the human kinetochore and localization domains of PLK1. CCAN = constitutive centromere associated network (inner kinetochore); KMH = KNL1 complex-MIS12 complex-NDC80 complex. Unstructured tail of CENP-U provides docking motifs for PLK1. PLK1 recruited to Thr98 after CDK1 phosphorylation promotes phosphorylation of Thr78 and recruitment of PLK1 to this site, a mechanism we define as relay priming. Unstructured region of KNL1 provides phosphorylated motifs (MELTs) that recruit BUB3:BUB1, which in turn recruits BUB3:BUBR1. CDK1 phosphorylation of BUB1 and BUBR1 promotes recruitment of PLK1, which in turn contributes to phosphorylation of short linear motifs (SLiM) on BUB1 and BUBR1 that recruit PP2A phosphatase. **C** Analytical SEC profiles and corresponding SDS-PAGE for phosphorylation-dependent and independent binding of the PBD and CENP-U. A, B, and C indicate the elution front of different interacting species. Equivalent species will be labelled with the same letter throughout the paper. MWM = molecular weight markers (in kDa). Source data are provided as a Source Data file. **D** Mass spectrometry analysis of phosphorylation of T98$^{CENP-U}$ (Motif 2; CDK1) under co-mixing or pre-phosphorylation conditions. Data from five replicates were imported

into GraphPad Prism 9.0 (GraphPad Software). Single dots represent data measurement for a single replicate, while the horizontal black bar represents the mean value. The error vertical black bar represents the 95% confidence interval of the mean. Statistical analysis was performed using a two-way ANOVA test (two-sided). Statistical output was represented with an asterisk-based system, only when $P \leq 0.0001$ (****), between conditions 2–4 and 3–5 or $P > 0.05$ (ns, not significant) between conditions 4–5 and 4–6. F-statistics and degrees of freedom were as follows: $F_{(12, 84)} = 1037$, $P < 0.0001$. Adjustments for multiple comparisons were performed using Tukey's multiple comparison test. Source data are provided as a Source Data file. **E** Mass spectrometry analysis of T78$^{CENP-U}$ (Motif 1; PLK1) under co-mixing or pre-phosphorylation. Data from five replicates were imported into GraphPad Prism 9.0 (GraphPad Software). Single dots represent data measurement for a single replicate, while the horizontal black bar represents the mean value. The error vertical black bar represents the 95% confidence interval of the mean. Statistical analysis was performed using a two-way ANOVA test. Symbol indication: ns (not significant) = $p > 0.05$, **** = $p \leq 0.0001$. F-statistics and degrees of freedom were as follows: $F_{(6, 56)} = 339.1$, $P < 0.0001$. Source data are provided as a Source Data file.

loop, are the main contacts[28]. Peptide binding is believed to restructure and stabilize the L2 loop, arguably through the action of the phosphate group, disrupting these contacts and releasing the kinase active site from its partial occlusion in the auto-inhibited form[28,30,31]. Through these mechanisms, phosphorylated targets of the PBD may unleash kinase activity to promote phosphorylation of substrate locally, near the target site, eliciting sequential phosphorylation[32–36].

Importantly, the PBD appears to be dispensable for most mitotic functions of PLK1, but completion of chromosome biorientation requires it[14,15,37,38]. Work on the kinetochore, a large protein assembly on chromosomes that promotes microtubule binding and biorientation[39,40], identified several kinetochore proteins as potential binding sites for the PLK1 PBD[20,21,41–57]. Two of these, CENP-U (a.k.a. PBIP1) and BUB1, respectively residing in the inner and outer kinetochores (Fig. 1B), emerged as master docking motifs for PLK1, as their suppression eliminated PLK1 recruitment to kinetochores altogether[16,58,59]. Thus, recruitment of PLK1 to the kinetochore and other cellular locales may be hierarchical, with few high-affinity master docking motifs (i.e., motifs most upstream in the pathway) being essential for initial targeting and persistent activation of PLK1. This may be followed by local phosphorylation of other targets, in some cases also leading to the creation of secondary, more transient docking sites. Recent work identifying a master docking motif in M18BP1 for recruitment of PLK1 to anaphase kinetochores is consistent with this concept[60,61].

Understanding the mechanism of PLK1 activation is urgent also in view of evidence that this kinase, at least in its more fundamental state, appears to be rather poorly active[62,63]. What defines master docking motifs, and whether self- or nonself-priming is preferentially involved in their generation, however, remains poorly understood. PLK1 recruitment to its putative master target motifs at the kinetochores, BUB1 and CENP-U, engaging both self- and nonself-priming mechanisms, exemplifies the potential molecular complexity of this problem. BUB1 binds PLK1 after CDK-mediated phosphorylation of T609[16,44,64–68]. BUB1 also recruits BUBR1, which itself interacts with PLK1 after CDK-mediated phosphorylation of T620[57,69,70]. Recruitment of PLK1, in turn, facilitates binding of PP2A phosphatase to the kinetochore[41,57,64,65,71–76].

CENP-U, a 416-residue protein, contains a C-terminal helical region (residues 250-416) playing a structural role in the inner kinetochore, and an unstructured N-terminal segment (residues 1-249) of uncertain function (Fig. 1A)[16,77]. PLK1 phosphorylates T78 of CENP-U and subsequently binds to the phosphorylated motif, an example of self-priming[19–21]. More recent evidence, however, suggested that T78 is inefficiently phosphorylated by PLK1 without prior phosphorylation of T98, a site within an S-T-P CDK1 substrate motif. The latter recruits

PLK1 to enhance subsequent phosphorylation of T78, in a relay priming mechanism (Fig. 1B)[16].

Initial biochemical experiments demonstrated that when T78 and T98 are concomitantly phosphorylated, two PBDs can dock side-a-side on CENP-U[16]. Dimerization can be harnessed to increase the effective binding affinity to a target and is also a mechanism frequently used in allosteric kinase activation[78,79]. Thus, the observations on CENP-U may reflect a strategy to promote local activation. Constellations of tandem PBD docking motifs seemingly similar to the one found in CENP-U are also present in BUB1, as well as in other PLK1 binding partners outside kinetochores[16]. Nonetheless, the generality of this mechanism is unknown, and so are the exact determinants of high-affinity binding of the PBD to target sequences, a property plausibly required for their ability to qualify as master PLK1 docking motifs. MAP205, an established biochemical inhibitor of PLK1 (without identified human orthologs), engages the phosphoresidue-binding site using a phospho-mimetic strategy. The MAP205 footprint on the PBD surface, however, is more extensive. It also includes a so-called cryptic pocket that was identified in the structure of the PBD with a peptide encompassing residues 71-79 of CENP-U[15,28,29,38,80,81].

Here, we revisit the mechanism of CDK priming on CENP-U and the generality of the PBD dimerization mechanism that emerged from initial analysis of the interaction of PLK1 with CENP-U in humans[16]. A crystal structure of two PBD domains bound to a CENP-U fragment encompassing both the pT78 and pT98 motifs reveals similarities and differences in the organization of the two bound PBDs. We confirm that initial CDK priming on T98 is required for efficient T78 relay-priming phosphorylation by PLK1. We show that PLK1 has unexpectedly high affinity for the motif encompassing pT78, approximately 100-fold stronger than for the pT98 motif. The high binding affinity for the pT78 motif reflects an extended interaction interface that occupies various docking stations on the PB1 and the PB2. On the basis of these observations, quantitative analyses, and additional structural modelling, we propose that PLK1 master docking motifs, whether generated by CDK1 or PLK1 itself, occupy a much greater area of the PBD surface than hitherto realised, and have several common structural features that distinguish them from transient sites. Collectively, our results have important implications for understanding PLK1 activation and function.

## Results

### pT98 primes phosphorylation of T78

We reconstituted in vitro a human PLK1:CENP-U complex with PBD fused N-terminally to maltose-binding-protein ($^{MBP}$PBD) and residues 58-114 of CENP-U fused N-terminally to MBP ($^{MBP}$CENP-U$^{58-114}$). In

analytical size-exclusion chromatography (SEC), both [MBP]CENP-U[58-114] and [MBP]PBD eluted in single peaks (Fig. 1C, yellow and blue lines). When mixed at a molar ratio of 2:1 (10 μM [MBP]PBD and 5 μM [MBP]CENP-U[58-114]), the two species underwent a leftward shift in elution volume and co-eluted (Fig. 1C; light grey line), suggesting that at the concentrations of these experiments, complex formation can occur also in the absence of phosphorylation. Using non-overlapping constructs encompassing the first (T78) or second (T98) phosphorylation site and neighbouring residues ([MBP]CENP-U[58-83] and [MBP]CENP-U[84-114], which we also refer to as Motif 1 and Motif 2, respectively), we determined that the PBD has residual binding affinity for unphosphorylated Motif 1, but not for unphosphorylated Motif 2 (Fig. S1A–C and see below).

To phosphorylate T78 and T98 of [MBP]CENP-U[58-114], we added catalytic amounts (typically 1:30 molar ratio) of active CDK1 and PLK1 kinase (see Methods). Successful phosphorylation was confirmed by Pro-Q™ Diamond staining, which specifically stains phosphorylated protein. We tested either 1) pre-phosphorylating [MBP]CENP-U[58-114] with both kinases before adding [MBP]PBD for complex formation (pre-phosphorylation method); or 2) mixing [MBP]CENP-U[58-114] and [MBP]PBD with both kinases for simultaneous phosphorylation and complex formation (co-mixing method). SEC analysis demonstrated that pre-phosphorylation (Fig. 1C; black trace) resulted in an apparently stoichiometric complex with a smaller elution volume relative to the complex generated by co-mixing (dark grey trace).

We used mass spectrometry (MS) to further assess CENP-U[58-114] phosphorylation potential in the co-mixing and pre-phosphorylation regimes. CDK1 (used at 100 nM against 5 μM substrate) was sufficient for robust phosphorylation of T98, regardless of whether pre-phosphorylation or co-mixing was used (Fig. 1D, conditions 4–5). Conversely, phosphorylation in the presence of PLK1 (also used at 100 nM; Fig. 1D, conditions 2–3) was either absent or below the detection limit. Under the specific conditions used, PLK1 needed pre-phosphorylation to phosphorylate CENP-U on T78, with modest phosphorylation in isolation and efficient phosphorylation when combined to CDK1 (Fig. 1E, conditions 3 and 7). We surmise that the significant binding affinity of the PBD for Motif 1, even before phosphorylation, suppresses phosphorylation of T78 when reactions are carried out in the presence of PBD in the co-mixing procedure. We do not attribute physiological significance to this observation, but discuss it as a relevant factor in the outcome of biochemical procedures to obtain the fully phosphorylated complex. As T98 is strongly phosphorylated even in absence of PLK1 (Fig. 1D, conditions 4-5), these results imply that efficient phosphorylation of T78[CENP-U] under the stringent conditions of this experiment only occurs when T98 is phosphorylated, i.e. when PLK1 and CDK1 are concomitantly present, confirming that CDK1 primes phosphorylation of T78 by PLK1 in a mechanism of relay priming. We note that despite a leucine at the −3 position, T78 has a less favourable histidine at the −2 position (see Introduction), rendering T78 a non-ideal PLK1 substrate and likely explaining why relay priming facilitates its phosphorylation.

## Crystallization and structure determination

We generated untagged versions of CENP-U[58-114] and the PBD for crystallization using pre-phosphorylation (see Methods). Crystals were obtained as described in Methods and had properties summarized in Table 1. Structure determination to a minimal Bragg spacing of 2.15 Å was obtained using Molecular Replacement as phasing method (Table 1, dataset 1). The CENP-U[58-114] peptide bridges two sequentially positioned PBDs, each of which bound to one of the two phospho-Thr (pT78 and pT98), with both phosphates showing unequivocal electron density (Figs. 2A, S2A–C). A short helical segment of CENP-U, the bridging helix (residues E84 to L93), connects the PBD-binding motifs, buried at the interface between the two PBDs. The latter do not establish direct contacts in the complex.

Motifs 1 and 2 bind their cognate PBDs in apparently similar ways (Fig. 2B, C). The side chains of the two phosphorylated residues, pT78[CENP-U] and pT98[CENP-U], are bound essentially identically through a pincer made by H538[PLK1] and K540[PLK1] [11,12], located in the second polo-box (PB2) of the PBD (Fig. 2B, C). R557[PLK1] (whose electron density is more evident at the binding site for pT78[CENP-U]) and a few water molecules complete the shell of interactions around the phosphate. In addition, both CENP-U motifs make extensive contacts with the PB1, most notably by inserting phenylalanine side chains (F71[CENP-U] and F87[CENP-U]) into a cavity formed by various aromatic side chains, including those of Y417[PLK1], Y421[PLK1], L478[PLK1], Y481[PLK1], and F482[PLK1] (Fig. 2B, C) and previously identified as the cryptic pocket (also identified as Tyr pocket)[15,81] for reasons that are clarified below. After mutating F87 to alanine, the stable complex of apparent 1:1 stoichiometry between [MBP]CENP-U[84-114] (encompassing Motif 2) and [MBP]PBD obtained after CDK1 phosphorylation was shifted to a greater elution volume, indicative of a loss of stability and confirming the importance of this interaction (Fig. S1D, E).

Despite different numbers of residues (six and ten, respectively), the distances between $C_\alpha$ atoms in the two F71[CENP-U]-pT78[CENP-U] and F87[CENP-U]-pT98[CENP-U] pairs are essentially identical (~19.5 Å) (Fig. 2D). Two consecutive prolines (P73-P74) promote a more extended main chain conformation between F71[CENP-U] and pT78[CENP-U], whereas the connection between F87[CENP-U] and pT98[CENP-U] is less extended and partly helical, explaining this. Thus, a common element of the two PLK1-binding motifs of CENP-U is that they bind the PBD as a staple, with an electrostatic interaction network around the phospho-threonine, and a hydrophobic interaction around a phenylalanine preceding the phospho-threonine, with an adaptable number of residues in between. Additional elements that instead diversify these motifs are discussed below.

## Quantitative analysis of the CENP-U:PBD interaction

Our observations so far are that phosphorylation of T78 by PLK1 is facilitated by prior phosphorylation of T98 by CDK1, and that two PBD domains can be bound to these residues at the same time, confirming and extending our previous results[16]. We previously hypothesized that dimerization of PLK1 on a target may give rise to a cooperative, more stable assembly, and that this could explain the ability of CENP-U to act as a master PLK1 docking site[16]. Our structure of two PBDs bound to CENP-U, however, did not reveal obvious evidence of cooperativity. The two PBD do not touch each other and are separated by the bridging helix. Therefore, we wanted to gain a more quantitative perspective of the mechanism of PLK1 binding to CENP-U.

For this, we measured binding thermodynamics and kinetics using biolayer interferometry (BLI) and isothermal titration calorimetry (ITC), obtaining association and dissociation rate constants ($k_{on}$ and $k_{off}$) and dissociation constant ($K_D$). We began by characterizing the interaction with Motif 1 by BLI. To obtain T78 phosphorylation in absence of T98, we increased the concentration of CENP-U[58-114] and active PLK1 kinase to bypass priming and phosphorylate T78 with only PLK1 (see Methods). Biotin-conjugated [MBP]PBD immobilized on the instrument's sensor tip surface bound to unphosphorylated [MBP]CENP-U[58-83] (only encompassing Motif 1) with a dissociation constant ($K_D$) of 10 μM (Table 2a, entry 1; Fig. S3A–L shows sensorgrams for all listed experiments). This relatively high basal affinity (i.e. in the absence of phosphorylation), and the fact that it arises from a very slow $k_{off}$, likely explains the ability of the PBD to inhibit T78 phosphorylation in the co-mixing regime. PLK1-phosphorylated [MBP]CENP-U[58-83], on the other hand, bound with a $K_D$ of 2.5 nM (Table 2a, entry 2). Thus, Motif 1 binds the PBD with very high binding affinity even in the absence of the neighbouring Motif 2. Mutations in Motif 1 of residues that bind at or near the cryptic pocket and complement the interaction of pT78 at the pincer,

**Table 1 | Data collection and refinement statistics**

| Data Collection | | | | | |
|---|---|---|---|---|---|
| Dataset | 1 | 2 | 3 | 4 | 5 |
| Constructs | CENP-U$^{58-114}$ PBD | CENP-U$^{39-114}$ PBD | CENP-U$^{58-114}$ PBD | CENP-U$^{58-114}$ PBD | PBD |
| Method | Pre-phosphorylation (PLK1 + CDK1) | Pre-phosphorylation (PLK1 + CDK1) | Co-mixing (PLK1 + CDK1) | Co-mixing (only CDK1) | --- |
| Beamline | SLS X10SA | ESRF ID23-2 | SLS X10SA | SLS X10SA | SLS X10SA |
| Temperature (K) | 100 | 100 | 100 | 100 | 100 |
| Wavelength (Å) | 1.000 | 0.873 | 1.000 | 1.000 | 1.000 |
| Space group | $P2_12_12$ | $P2_12_12$ | $P2_12_12$ | $P2_12_12$ | $P2_12_12$ |
| Unit cell parameters (Å) | 84.89 (a) | 83.79 (a) | 84.67 (a) | 84.94 (a) | 76.70 (a) |
| | 135.09 (b) | 134.17 (b) | 135.40 (b) | 136.42 (b) | 98.57 (b) |
| | 60.95 (c) | 60.89 (c) | 60.36 (c) | 60.29 (c) | 33.23 (c) |
| | 90° (α) | 90° (α) | 90° (α) | 90° (α) | 90° (α) |
| | 90° (β) | 90° (β) | 90° (β) | 90° (β) | 90° (β) |
| | 90° (γ) | 90° (γ) | 90° (γ) | 90° (γ) | 90° (γ) |
| Resolution range (Å) | 42.44–2.15 | 45.09–2.00 | 45.05–2.00 | 36.3–2.10 | 49.28–1.60 |
| | (2.20–2.15) | (2.04–2.00) | (2.04–2.00) | (2.15–2.10) | (1.65–1.60) |
| No. of measured reflections | 526,136 (37,950) | 528,635 (30,761) | 645,757 (38,741) | 555,290 (38,084) | 444,296 (37,792) |
| No. of unique reflections | 38,912 (2723) | 46,806 (2711) | 47,668 (2738) | 41,642 (2725) | 34122 (2775) |
| Multiplicity (redundancy) | 13.5 (13.9) | 11.3 (11.3) | 13.5 (14.1) | 13.3 (14.0) | 13.0 (13.6) |
| Completeness (%) | 99.97 (100.00) | 99.17 (98.98) | 99.95 (99.96) | 99.83 (99.45) | 99.91 (99.82) |
| Mean $I_O/\sigma(I_O)$ | 15.69 (1.17) | 8.62 (1.21) | 19.73 (1.15) | 17.57 (1.17) | 9.84 (1.20) |
| $R_{merge}$ [a] (%) | 8.2 (188.0) | 16.5 (235.0) | 6.8 (212.7) | 8.1 (212.7) | 14.2 (282.6) |
| $CC_{1/2}$ | 0.999 (0.572) | 0.997 (0.399) | 1 (0.554) | 1 (0.594) | 0.999 (0.238) |
| Refinement | | | | | |
| Reflections used in refinement | 38,901 (2723) | 46,796 (2711) | 47,649 (2736) | 41,628 (2725) | 34,104 (2774) |
| Reflections used for R-free | 1945 (136) | 2340 (136) | 2381 (137) | 2080 (137) | 1706 (139) |
| $R_{work}$ [b] (%) | 21.7 (33.5) | 20.7 (35.3) | 21.1 (41.4) | 21.0 (36.1) | 19.73 (41.13) |
| $R_{free}$ [c] (%) | 25.0 (34.6) | 24.1 (37.3) | 24.2 (43.5) | 24.3 (38.6) | 22.85 (46.14) |
| Number of non-hydrogen atoms | 4069 | 4159 | 4110 | 4108 | 2020 |
| macromolecules | 3979 | 4037 | 3939 | 3945 | 1914 |
| ligands | 38 | 13 | 66 | 52 | 17 |
| solvent | 52 | 109 | 105 | 111 | 89 |
| Protein residues | 485 | 496 | 481 | 480 | 229 |
| R.m.s.d. bond length (Å) | 0.002 | 0.005 | 0.003 | 0.002 | 0.007 |
| R.m.s.d. bond angles (°) | 0.53 | 0.72 | 0.64 | 0.58 | 0.87 |
| Ramachandran plot favoured region (%) | 95.98 | 96.68 | 97.67 | 97.66 | 96.48 |
| Ramachandran plot allowed region (%) | 4.02 | 3.32 | 2.33 | 2.34 | 3.52 |
| Ramachandran plot outliers (%) | 0.00 | 0.00 | 0.00 | 0.00 | 0.00 |
| Clashscore | 3.69 | 3.37 | 3.39 | 4.78 | 3.64 |
| Average B-factor (Å$^2$) | 67.98 | 53.34 | 57.00 | 61.36 | 32.81 |
| PDB ID | 9FJH | 9FJJ | 9FJG | 9FJI | 9QMO |

a $R_{merge} = \Sigma_{hkl} \Sigma_i [|I_i(hkl) - <I(hkl)>| / \Sigma_{hkl} \Sigma_i I_i(hkl)]$, where $I_i(hkl)$ is the intensity value of the *i*th measurement of reflection *hkl*, *<I(hkl)>* is the mean value of $I_i(hkl)$ for all *i* measurements. b $R_{work} = \Sigma_{hkl} ||F_{obs}| - |F_{calc}|| / \Sigma_{hkl} |F_{obs}|$, where $F_{obs}$ and $F_{calc}$ are respectively the observed and calculated structure factors. c $R_{free}$ is the free $R_{work}$ for the 5% of reflections that were excluded from the refinement process.

including Y68A and F71A, severely affected binding of $^{MBP}$PBD to Motif 1 (Table 2a, entry 3-4). An $^{MBP}$CENP-U$^{58-114}$ construct carrying mutations impairing Motif 2, and phosphorylated with only PLK1, had a dissociation constant very similar to that observed with the phosphorylated Motif 1 (Table 2a, entry 5). Unlike Motif 1, unphosphorylated Motif 2 did not bind the PBD in BLI experiments, while a peptide phosphorylated on T98 bound with a $K_D$ of ~400 nM (Table 2a, entries 6-7).

To investigate the putative cooperativity of the interaction of PBDs with CENP-U$^{58-114}$, we biotin-conjugated and immobilized the latter and measured binding affinity to singly and doubly phosphorylated CENP-U. We reasoned that cooperative binding should manifest itself as a noticeable increase of binding affinity under conditions allowing the occupation of both binding sites by the PBD. We initially immobilized $^{MBP}$CENP-U$^{58-114/pT78}$ and compared the binding affinity of the PBD with that of full length PLK1 (PLK1$^{FL}$; Table 2b, entries 8-9). Fitting of binding kinetics indicated that $^{MBP}$CENP-U$^{58-114/pT78}$ contains a single binding site (centred around Motif 1). This is consistent with the observation that the unphosphorylated Motif 2 does not bind the PBD (Table 2a, entry 6).

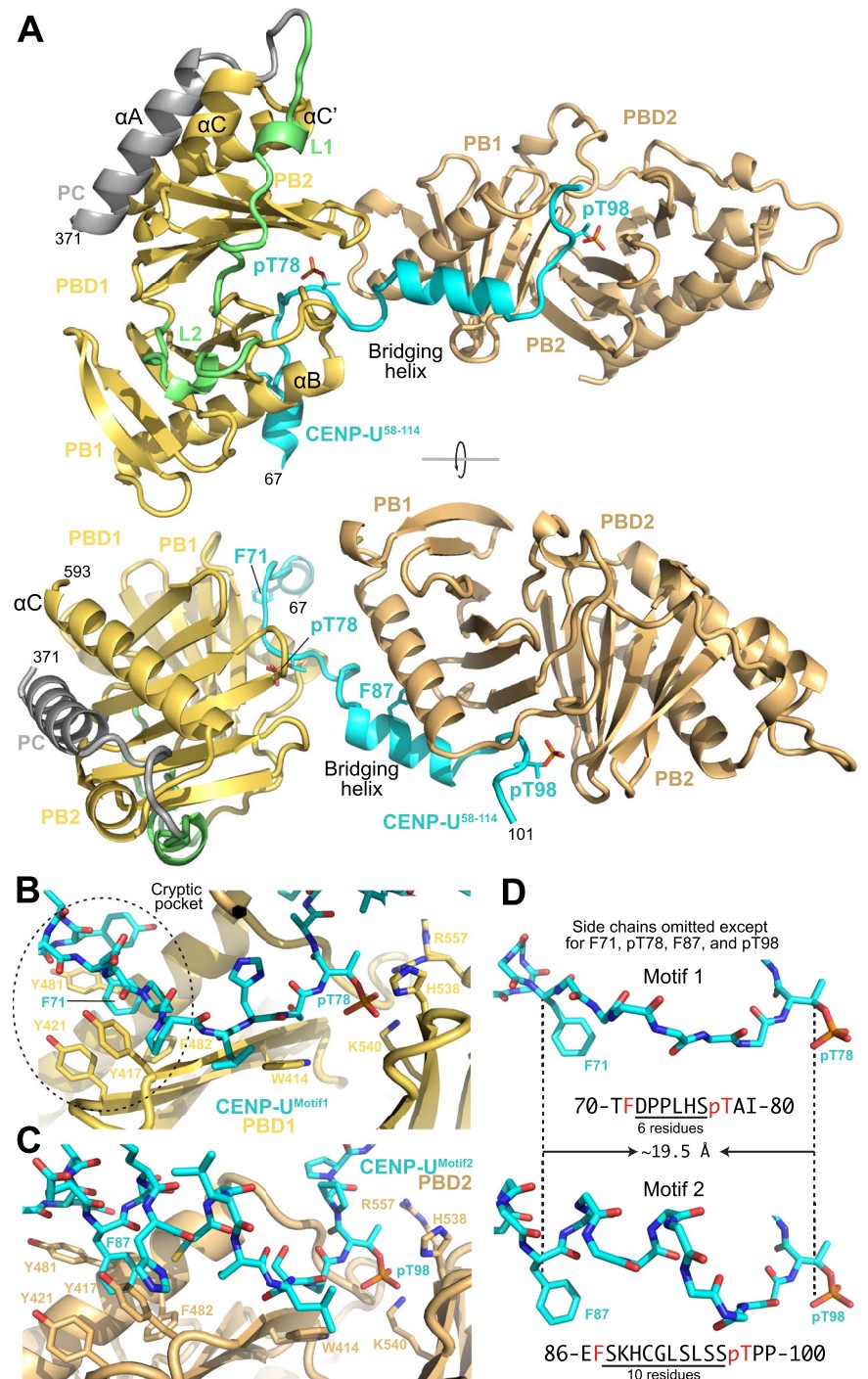

**Fig. 2 | Structural basis for CENP-U binding by PLK1. A** Two rotated cartoon models of the crystal structure from Dataset 1 of a trimeric complex formed by CENP-U[58-114] with both T78[CENP-U] and T98[CENP-U] phosphorylated (cyan) and two bound PBDs (identified as PBD1 and PBD2 and shown in yellow orange and wheat, respectively). Key residues are indicated in stick representation. Colouring as per Fig. 1A. Helices were named as in ref. 12. **B** Closeup of the pT78 motif represented in sticks against a cartoon of PBD1 with key residues highlighted in stick representation. The position of the cryptic pocket hosting F71 (and F87 in (**C**)) is highlighted. **C** As in (**B**) for pT98. **D** Comparison of Motif 1 and Motif 2 with Phe (F) and pT representing two pins of a staple inserted respectively into the hydrophobic cryptic pocket and the electrostatic pincer of PBD1. The distance between corresponding Cα atoms is essentially identical.

Next, we repeated the binding experiments on immobilized CENP-U[58-114/pT98] (Table 2b, entries 10-11). In these two measurements, analysis of binding parameters suggested the existence of a primary, higher-affinity binding site ($K_D1$) and of a secondary, lower-affinity binding site ($K_D2$). In view of results in Table 2a, which were obtained with the immobilized PBD, $K_D1$ likely reflects binding to the phosphorylated Motif 2, while $K_D2$ reflects binding to the unphosphorylated Motif 1. Finally, we immobilized doubly phosphorylated [MBP]CENP-U[58-114/pT78/pT98] and once more, monitored binding of the PBD (Table 2b, entries 12-13). Also in this case, analysis of binding parameters suggested the existence of a primary, higher affinity binding site ($K_D1$, presumably reflecting binding to Motif 1) and of a secondary, lower affinity binding site ($K_D2$, presumably reflecting binding to Motif 2).

**Table 2 | BLI analysis quantified binding kinetic constants**

| Entry | Immobilized Ligand | Analyte | $K_D$ (M) | $k_{on}$ (1/Ms) | $k_{off}$ (1/s) |
|---|---|---|---|---|---|
| a \| Binding parameters of PBD to variants of CENP-U (unmodified or phosphorylated) | | | | | |
| 1 | $^{MBP}$PBD | $^{MBP}$CENP-U$^{58-83}$ | $1.0 \times 10^{-5}$ | $2.8 \times 10^{1}$ | $2.8 \times 10^{-4}$ |
| 2 | | $^{MBP}$CENP-U$^{58-83/pT78}$ | $2.5 \times 10^{-9}$ | $1.4 \times 10^{5}$ | $3.4 \times 10^{-4}$ |
| 3 | | $^{MBP}$CENP-U$^{58-83/F71A/pT78}$ | $8.8 \times 10^{-8}$ | $8.7 \times 10^{4}$ | $7.7 \times 10^{-3}$ |
| 4 | | $^{MBP}$CENP-U$^{58-83/Y68A/F71A/pT78}$ | $2.3 \times 10^{-7}$ | $4.7 \times 10^{4}$ | $1.1 \times 10^{-2}$ |
| 5 | | $^{MBP}$CENP-U$^{58-114/pT78/F87A/T98V}$ | $7.2 \times 10^{-9}$ | $1.4 \times 10^{5}$ | $1.0 \times 10^{-3}$ |
| 6 | | $^{MBP}$CENP-U$^{84-114}$ | N.B.$^a$ | --- | --- |
| 7 | | $^{MBP}$CENP-U$^{84-114/pT98}$ | $4.0 \times 10^{-7}$ | $7.1 \times 10^{3}$ | $2.8 \times 10^{-3}$ |
| b \| Binding parameters of PBD or PLK1$^{FL}$ to variants of CENP-U (unmodified or phosphorylated) | | | | | |
| 8 | $^{MBP}$CENP-U$^{58-114/pT78}$ | $^{MBP}$PBD | $1.9 \times 10^{-9}$ | $2.8 \times 10^{5}$ | $5.2 \times 10^{-4}$ |
| 9 | | $^{6His}$PLK1$^{FL}$ | $2.2 \times 10^{-8}$ | $1.5 \times 10^{5}$ | $3.3 \times 10^{-3}$ |
| 10 | $^{MBP}$CENP-U$^{58-114/pT98}$ | $^{MBP}$PBD | $1.6 \times 10^{-8}$ ($K_D$1) | $4.5 \times 10^{4}$ | $7.1 \times 10^{-4}$ |
| | | | $4.0 \times 10^{-6}$ ($K_D$2) | $7.4 \times 10^{3}$ | $2.9 \times 10^{-2}$ |
| 11 | | $^{6His}$PLK1$^{FL}$ | $4.2 \times 10^{-6}$ ($K_D$1) | $2.3 \times 10^{3}$ | $1.0 \times 10^{-2}$ |
| | | | $1.1 \times 10^{-5}$ ($K_D$2) | $5.6 \times 10^{4}$ | $6.0 \times 10^{-1}$ |
| 12 | $^{MBP}$CENP-U$^{58-114/pT78/pT98}$ | $^{MBP}$PBD | $2.7 \times 10^{-9}$ ($K_D$1) | $4.5 \times 10^{4}$ | $1.2 \times 10^{-4}$ |
| | | | $1.0 \times 10^{-6}$ ($K_D$2) | $1.8 \times 10^{4}$ | $1.8 \times 10^{-2}$ |
| 13 | | $^{6His}$PLK1$^{FL}$ | $1.0 \times 10^{-8}$ ($K_D$1) | $1.7 \times 10^{5}$ | $1.8 \times 10^{-3}$ |
| | | | $1.6 \times 10^{-6}$ ($K_D$2) | $9.4 \times 10^{3}$ | $1.5 \times 10^{-2}$ |

*a*) N.B., no binding.

Using ITC, we measured the binding affinity of the PBD for singly phosphorylated pT78 and pT98 peptides. The ITC measurements were in excellent agreement to the BLI measurements, identifying a $K_D$ of ~380 nM for the pT98 peptide, and of ~1.9 nM for the pT78 peptide (Fig. S3M–O). These measurements, obtained with an approach orthogonal to BLI (ITC is performed in solution and measures the concentration of binding product at equilibrium, rather than association and dissociation kinetics), confirm that pT78 provides a very high-affinity binding site for the PLK1 PBD.

Thus, collectively, several patterns emerge with clarity from these analyses. First, the PBD binds to pT78 with approximately 100-fold higher affinity than to pT98. Second, the PBD binds CENP-U with systematically higher affinity than the PLK1$^{FL}$, quantifiable in approximately one order of magnitude difference in the dissociation constant. We confirmed this conclusion by using orthogonal approaches, including SEC (Fig. S4A–D) and SEC-Multi Angle Light Scattering (SEC-MALS; Fig. S5A–G). We can only speculate on the causes of this difference in binding affinity, but suspect it may reflect a) the larger hydrodynamic radius of PLK1$^{FL}$ relative to the PBD, which in turn reflects on a larger $k_{off}$; and b) an energy penalty associated with the disruption of intramolecular communication between the PBD and the kinase domain. Third, the presence of two PBDs on CENP-U does not appear to have a strong positive cooperative effect. If anything, presence of two phosphorylation sites may even appear to slightly increase the dissociation constants, i.e. act anti-cooperatively. From this, we conclude that docking of two PBDs on CENP-U, while clearly happening, may not be a crucial factor in the creation of a PLK1 master docking site. Rather, our data suggest that CDK1 phosphorylates CENP-U on Thr98 to attract PLK1 to an otherwise poorly phosphorylated site, Thr78, to which PLK1 then binds very tightly, in line with the idea that pThr78 may represent a master PLK1 docking site.

### Additional interactions of CENP-U with the PBD

As discussed in the Introduction, the precise molecular determinants of master docking sites for PLK1 recruitment remain unclear. To date, structural work has captured PBD interactions of short phosphopeptides binding exclusively in the vicinity of the phosphoresidue binding pocket (see for instance[12,81–84]). In addition, structures of residues 71-79 of CENP-U in complex with the PBD identified an additional point of contact at the cryptic pocket[15,81]. The structure calculated from Dataset 1 and our subsequent biochemical experiments add significantly to the picture. First, it appears that occupation of the cryptic pocket is more common than hitherto appreciated, as we see it not only with the pT78-containing motif, but also with the pT98-containing motif. Our BLI experiments showed reduced binding affinity when F71, which occupies the cryptic pocket, was mutated (Table 2). This is in line with previous experiments with 8- or 9-residue synthetic peptides encompassing the pT78 residue, and reporting 5- to 9-fold increases in binding affinity when this pocket becomes engaged with phenylalanine[81].

Thus, occupation of the cryptic pocket leads to an increase of binding affinity relative to peptides with shorter N-terminal extensions occupying only the pincer site. Of note, the PBD binding affinity of the 9-residue synthetic peptide of CENP-U encompassing pT78 (0.25 μM)[81] is among the highest measured so far. When binding is limited to the pincer pocket, measured binding affinities are in the micromolar range[12,71,81–84]. In our experiments comprising two orthogonal approaches (BI and ITC, Table 2 and Fig. S3), the pT98-containing motif bound the PBD with a dissociation constant of 0.4 μM (Table 2), similar to the 9-residue pT78 synthetic peptide[81]. Instead, a pT78-containing motif with a longer N-terminal extension binds the PBD with at least 100-fold higher binding affinity. This strongly suggests that additional interactions between Motif 1 and the PBD must be at stake, even beyond the cryptic pocket.

A comparison of the two PBD:CENP-U sub-structures (Fig. 3A) provides possible hints towards the source of this additional binding affinity. The CENP-U chain arriving from PBD1 contacts PBD2 immediately before the cryptic pocket (at Phe87). Conversely, there is extensive unexplored surface of PBD1 to which the CENP-U region preceding Phe71 could bind. Indeed, Tyr68 is an important determinant to the overall binding affinity of the CENP-U:PBD complex (Table 2), thus indicating that even residues preceding Phe71 contribute to increase the binding affinity of this motif for the PBD.

### Additional determinants of master binding sites

To further investigate this question, we extended the CENP-U construct on its N-terminal end by generating CENP-U$^{39-114}$ and reconstituted its interaction with the PBD. In SEC experiments, the peak of the PBD:CENP-

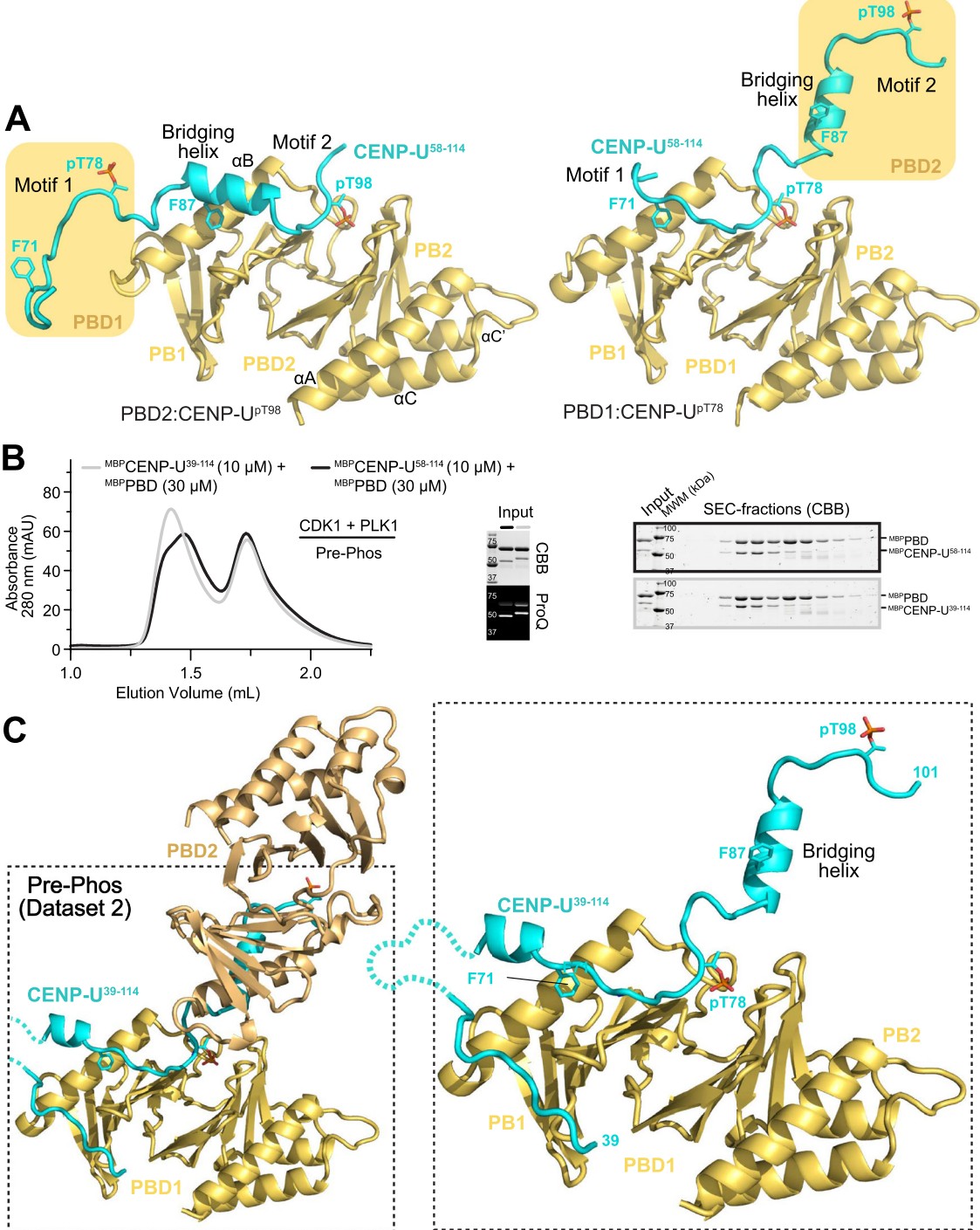

**Fig. 3 | More extended interaction of CENP-U³⁹⁻¹¹⁴. A** Comparison of the interaction of CENP-U⁵⁸⁻¹¹⁴ with PBD1 and PBD2 (right and left, respectively) suggests the region N-terminal to Motif 1 might interact with the exposed face of the PBD, like in the structure with MAP205 (4J7B). **B** Comparison by SEC and SDS-PAGE of PBD complexes with CENP-U⁵⁸⁻¹¹⁴ or CENP-U³⁹⁻¹¹⁴ generated by pre-phosphorylation with CDK1 and PLK1. The ProQ™ reagent interacts with phosphorylated species (see

Methods). CBB = Coomassie Brilliant Blue. MWM = molecular weight markers (in kDa). Source data are provided as a Source Data file. **C** Cartoon model of the crystal structure calculated from dataset 2 shows a very similar arrangement to that shown in Fig. 2A (and identical colouring scheme), but with an N-terminal extension running transversally between the two PBs.

U³⁹⁻¹¹⁴ complex eluted similarly to that of the PBD:CENP-U⁵⁸⁻¹¹⁴ complex, but appeared more homogeneous (Figs. 3B, S6A). The PBD:CENP-U³⁹⁻¹¹⁴ complex generated by pre-phosphorylation crystallized under conditions similar to those of the PBD:CENP-U⁵⁸⁻¹¹⁴ complex, and we determined its 2.0 Å resolution crystal structure (Table 1, dataset 2; Figs. 3C, S6B, C). As expected, the structure of PBD:CENP-U³⁹⁻¹¹⁴ is very similar to that of the PBD:CENP-U⁵⁸⁻¹¹⁴ complex, with two PBDs bound

consecutively to the CENP-U chain (Fig. 3C). The main difference is that there is clear density for several residues in the N-terminal extension of CENP-U that were not included in the PBD:CENP-U⁵⁸⁻¹¹⁴ complex, including a segment from residues 39 to 47 (Figs. 3C, S6B, C). Other parts of the segment were disordered and not visible in the electron density.

How CENP-U³⁹⁻¹¹⁴ binds the PBD is reminiscent of how MAP205, a PLK1 inhibitor, interacts with the PBD in a structure encompassing near

full-length PLK1 (from the zebrafish *Danio rerio*) reconstituted from two separate halves (roughly comprising the KD and part of the IDL on one side, and the PC and PBD domain on the other; see Fig. 6B), as well as residues 276–325 of MAP205 (from the fruit fly *Drosophila melanogaster*; PDB ID 4J7B[28]) (Fig. 4A). F304 of MAP205 (equivalent to F71 of CENP-U) positions itself in the cryptic pocket, while two acidic residues (E314 and E316) interact in the pincer pocket, replacing the canonical phosphorylated residue. Closer to its N-terminus, the MAP205 chain meanders over the surface of the PBD, interacting with additional surface features (Fig. 4B). The first is a highly conserved hydrophobic pocket lined by the side chains of Y412 (shared with the cryptic pocket), K465, and K466 (respectively equivalent to Y421, K474, and K475 of human PLK1), and which we refer to as the YKK pocket (Fig. 4B–D). The side chains of M293, I296, and V298 of MAP205 occupy this pocket, in a way that relates to how I41, V43, and F46 of CENP-U interact with the same pocket (Fig. 4C, D).

We tested the effects of mutational perturbations of the PBD by introducing alanine mutations in the pincer, cryptic, and cryptic-YKK pockets, and in proximity of the pseudo-pincer and pseudo-cryptic pockets. We then monitored mitotic kinetochore localization of wild-type and mutant mNeonGreen (mNG)-tagged PLK1 transgenes (Fig. 4E, quantified in Fig. 4F, G). Mutation of the pincer (H538A-K540M, therefore defined as AM mutant) ablated kinetochore recruitment of PLK1. Mutation of Y481, in the cryptic pocket, strongly affected kinetochore recruitment of mNG-PLK1. Mutation of Y421, whose side chain participates in both the cryptic and YKK-pockets, led to a level of impairment of kinetochore recruitment that was comparable to that resulting from mutation of the pincer.

## Conformational regulation of PLK1 through ligand binding

In addition to the pincer, cryptic, and YKK pockets, the MAP205 chain approaches two additional pockets positioned roughly pseudo-symmetrically to the pincer and cryptic pockets, and to which we therefore refer to as pseudo-pincer and pseudo-cryptic pocket (Figs. 4B and S6D). The pseudo-cryptic pocket is not occupied in the MAP205 structure (Figs. 4A, B), but AlphaFold 3 (AF)[85] modelling of a longer fragment of MAP205 (starting at residue 270 instead than at 276, the construct used in crystallization of 4J7B) predicts that the pseudo-cryptic pocket may also become occupied (Fig. 5A). AF modelling of PBD complexes of BUBR1, BUB1, and PRC1 (the latter in the sequence preceding and including T602 – using Uniprot isoform 4 as sequence reference – a known target of PLK1 phosphorylation[22,86]), predicted binding modes very similar to that of MAP205, with concomitant occupation of pincer, cryptic pocket, pseudo-pincer, and pseudo-cryptic pocket, despite relatively loose sequence constraints (Figs. 5B–E, S7A–C; a prediction of the PBD:CENP-U complex is also discussed in the context of Figs. S7D–F). Mutations of Y582 to Ser, in the pseudo-cryptic box, or replacement of S593 (on the predicted path of the chain) with a bulky side chain (Trp) did not appear to overtly affect mitotic kinetochore localization of the resulting mutants (Fig. S8A–C), probably because interactions at the pincer, cryptic, and YKK pockets are sufficient for strong kinetochore localization.

Collectively the experimental and predicted structures indicate that the PLK1 inhibitor MAP205 and the PLK1 activators BUB1, BUBR1, PRC1, and CENP-U all form extensive and rather similar contacts with the PBD. As the PBD is known to dampen PLK1 activity when engaged intra-molecularly with the kinase domain (see below), how these PBD ligands achieve opposite effects on PLK1 kinase activity remains unclear. In SEC assays, the isolated PBD and kinase domains of human PLK1 bound weakly (Fig. 5F). MAP205, by binding the PBD, strongly promoted stabilization of a ternary complex with the kinase domain, indicated by co-elution of all subunits (Fig. 5F), as observed previously[28]. BUB1 and CENP-U, on the other hand, destabilized the interaction of the PBD and kinase domains, partly in response to their phosphorylation state. Specifically, unphosphorylated BUB1 did not

interact with the PBD and did not interfere with the PBD-kinase interaction, but CDK1-mediated phosphorylation of BUB1 strongly promoted binding to the PBD, causing dissociation of the kinase domain (Figs. 5G, S8D). CENP-U[1-96], encompassing Motif 1, bound the PBD even before phosphorylation by PLK1 on T78. The interaction was reinforced after phosphorylation, and also in this case it caused dissociation of the kinase domain from the PBD:phosphopeptide complex (Fig. 5H). Thus, despite the similar engagement of the PBD by docking sites predicted by AlphaFold, MAP205 stabilizes a ternary complex with the kinase and PBD domain, whereas CENP-U and BUB1 disrupt the interaction of the PBD with the kinase domain, as expected for these bona fide activators of PLK1. These experiments also demonstrate that CENP-U is a very potent PBD binder, even in comparison to BUB1.

## Open and closed conformations of PLK1

The PBD is known to dampen PLK1 kinase activity[8,26–30,87]. The mechanism of opening and activation of the kinase domain by phosphorylated ligands bound to the PBD, however, remains unclear. To shed light on this problem, we predicted a structure of full length human PLK1, with or without bound ATP or T210 phosphorylation, with AF3. These essentially identical predictions showed a closed conformation, with the PBD docked tightly against the kinase domain (Figs. 6A, S9A–E). In the already mentioned structure of the PLK1:MAP205 complex (4J7B)[28], the two separate halves of PLK1 in the 4J7B structure are reciprocally arranged essentially identically to full-length PLK1 in the AF prediction (Fig. 6B), confirming that AF predicts a closed conformation for full-length PLK1 in the of PBD ligands.

None of the AF structures with bound phosphopeptides discussed above predicts direct clashes of the extended target peptide with the KD (not shown), suggesting an allosteric mechanism of kinase activation by phosphorylated target motifs. In this regard, the L2 loop of the PLK1 PBD, which connects the PB1 and PB2, emerges as a crucial potential conformational sensor. The base of this loop, at the exit from the αB helix, is juxtaposed to the pincer pocket (Fig. 6C), and, therefore ideally suited to sense binding of phosphopeptide targets. In the closed structure predicted by AF, contacts between the PBD and kinase domain are indirect and mediated by the interdomain-linker (IDL) and the L1 loop. The L2 loop does not directly contact the kinase domain but is adjacent to the IDL. The apex of L2 (depicted in lime) interacts with the first segment of the IDL, near the kinase hinge domain (Fig. 6A, C). In the inhibited PLK1:MAP205 complex, the L2 apex and IDL are in the same position as in the AF prediction of full-length PLK1 (Fig. 6B, D).

Conversely, in our structures of the PBD bound to doubly phosphorylated CENP-U (Table 1, datasets 1 and 2), the L2 loop adopts a distinct conformation that is clearly discernible in the available electron density, with weak density only for a few residues centred on Arg500 (Fig. 6E, S10A–E). Overall, there is a very significant rotation of the base of the L2 loop, in such a way that in the CENP-U-bound complex the apex of the L2 loop occupies a distinct position that overlaps with that adopted by the IDL in the closed conformation (Fig. 6E). Thus, relocation of the L2 loop upon phosphopeptide binding may cause the extrusion of the IDL and of the associated kinase domain into an unrestrained, active conformation (as exemplified by the arrows in Fig. 6E). A helpful beacon associated with the transition from the closed to the open conformation is the side chain of Ile497, for which there is excellent electron density, and whose position is marked for comparison in the context of the AF model and of the model calculated from dataset 2 (Fig. 6C, E, red arrow).

## Stabilization and destabilization of the base of the L2 loop

The base of the L2 loop is invisible in the MAP205 structure (Fig. 6D), likely because it is displaced by MAP205. The side chain of L315[MAP205], in the middle of the phosphomimetic 314-E-L-E-316 motif of MAP205,

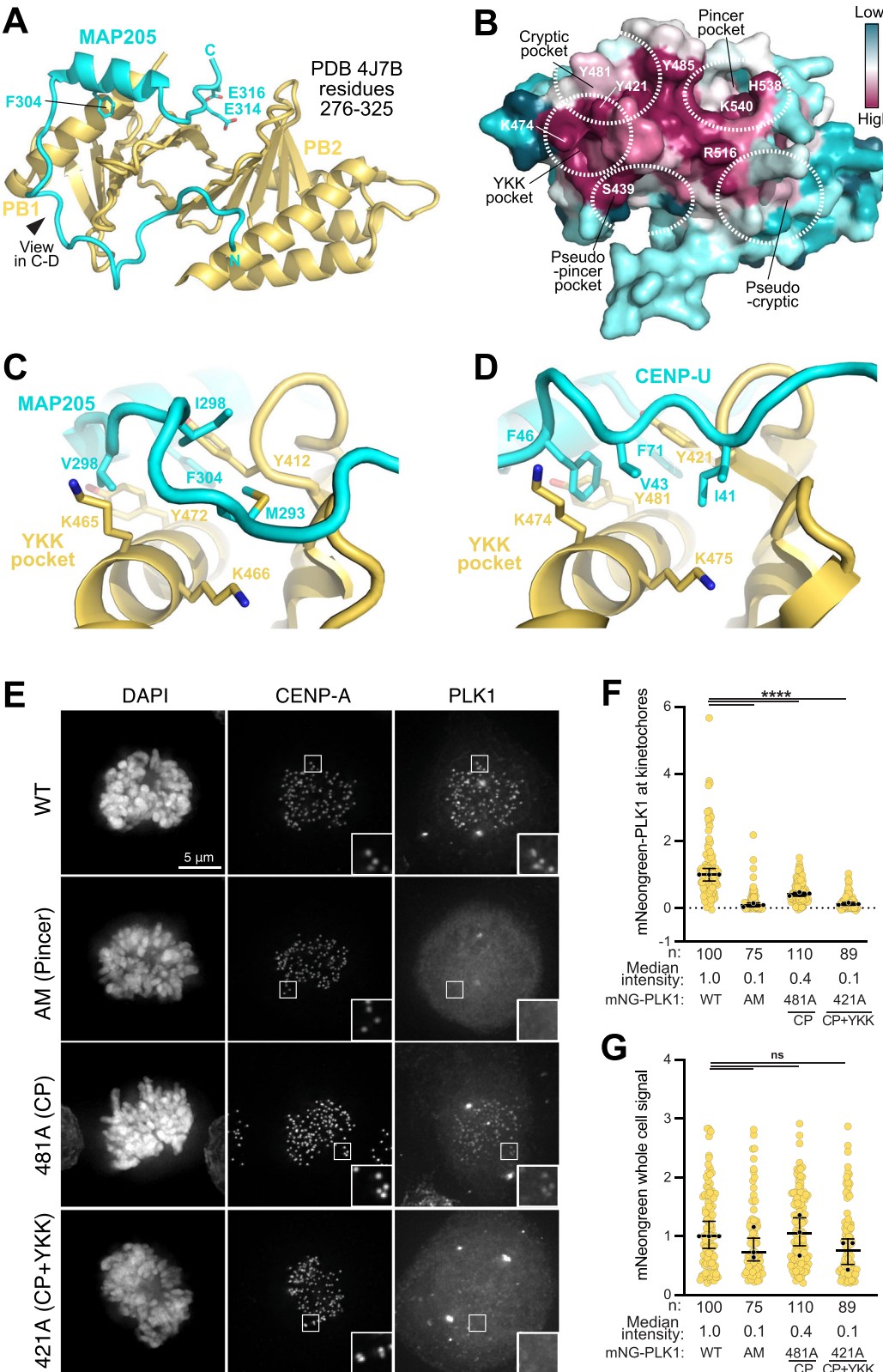

occupies the position of PLK1$^{L490}$ in the AF model of full-length PLK1 (compare panels C, D of Fig. 6). Thus, the likely mechanism through which MAP205 stabilizes the closed conformation is by forcing the base of the L2 loop into a conformation that is incompatible with opening.

These considerations also raise the question whether the con-formational change at the base of the L2 loop is strictly related to the presence of a phosphate group on target phosphopeptide. To address this, we raised crystals of mono-phosphorylated PBD:CENP-U$^{58-114}$ complex (with unphosphorylated T78 and phosphorylated T98) using the co-mixing procedure with both kinases or with only CDK1. For both procedures, crystals grew under largely similar conditions to those of the fully phosphorylated complex obtained by pre-phosphorylation (Table 1, datasets 3 and 4). Structures calculated with datasets 3 and 4

**Fig. 4 | Role of additional PBD pockets in the interaction with targets. A** Crystal structure of the MAP205 peptide of *Drosophila melanogaster* bound to the Zebrafish PBD, extracted from PDB ID 4J7B[28]. The pincer and cryptic pockets are occupied, but the N-terminal part of the MAP205 chain explores additional elements of the PBD. **B** Conservation on the PBD surface, calculated from over fifty distant PLK1 sequences from evolutionarily distant organisms, shown with the same orientation of panel A. Various pockets discussed in the text are shown. **C, D** Equivalent hydrophobic interactions of the MAP205 and CENP-U chains with the YKK pocket. The approximate view is that indicated by a arrowhead in (**A**). **E** Subcellular localization of an mNeonGreen-fused PLK1 transgene and the indicated mutants in mitotic HeLa cells. The first row with the wild type transgene is also shown in Fig. S8A as samples are part of the same experiment.

**F, G** Quantification of the experiments in panel E. The black bars represent the median of three replicates, and the vertical bar 95% confidence interval. Yellow dots represent either kinetochore or whole cell mNeongreen signal relative to the control condition (median set to 1). Black dots are medians of the single replicate. Statistical analysis was performed using the Kruskal-Wallis test as described in Methods. To convert *P* values into an asterisk-based significance system, we used the default GraphPad Prism convention: not significant (ns) $P > 0.05$; *$P \le 0.05$ but >0.01; **$P \le 0.01$ but > 0.001; ***$P \le 0.001$ but >0.0001; and ****$P \le 0.0001$. The exact *P* values for pairs of conditions in (**F**) were: WT vs. H538A/K540M $\le 0.0001$; WT vs. Y421A $\le 0.0001$; WT vs. Y481A $\le 0.0001$. The exact *P* values for pairs of conditions in (**G**) were: WT vs. H538A/K540M > 0.05; WT vs. Y421A > 0.05, WT vs. Y481A > 0.05. Source data are provided as a Source Data file.

were largely similar to that of the doubly phosphorylated complex (datasets 1 and 2). However, lack of the corresponding electron density indicated unequivocally that T78^CENP-U was not phosphorylated (Fig. 7A–C), while T98^CENP-U appeared, as expected, robustly phosphorylated (not shown).

Comparison of the organization of the L2 loop in the dataset 3 and 4 structures indicated that in all cases it adopts conformations that are much more closely related to the open state already observed with datasets 1 and 2 than to the closed, inhibited state (Fig. 7D). A cautionary note is that anomalous X-ray scattering (AXRS) detected an unbound free phosphate ion near the position normally occupied by the phosphate group of pT78 in structures from datasets 3 and 4 (indicated by an asterisk in Fig. 7B, C). While possible, it is unlikely that this would be a sufficient trigger to stabilize a particular conformation of the L2 loop (see also Discussion). Thus, collectively, this comparison does not appear to support a univocal association between the binding of pThr/pSer in the pincer pocket and the conformation of the L2 loop. This was further confirmed by the determination of a crystal structure of the apo (unbound) PBD domain (Table 1, dataset 5). While not entirely visible, the L2 loop in the apo-PBD structure adopts a conformation rather similar to that of the liganded PBDs, clustering in an open group that is clearly distinct from that expected for the closed assembly (Figs. 6F, 7D). Incidentally, the cryptic pocket is occluded in apo PBD, while it is available and occupied by peptide in all eight PBD:CENP-U sub-complexes in our datasets 1 to 4, justifying the name cryptic for this pocket (Fig. S9F-H).

## The Allopole-A binding site

In the open conformation of the L2 loop, the side chain of Ile497 relocates near a pocket formed by Trp410 and Phe559 (Figs. 6C–E, 7E). This pocket is the target of a small-molecule allosteric inhibitor of PLK1, Allopole-A, proposed to inhibit ligand binding by dislodging the L2 loop from its position in open structures[31]. In isolated PBDs, Ile497 packs against the Trp410 and Phe559 side chains both in peptide-bound and in the apo form (compare Fig. 6E, F). The side chain of Phe559 adopts two different conformations in our structures, which we refer to as Phe IN and Phe OUT (Fig. 7E). We observe fully Phe IN and Phe OUT conformation in PBD1 of the doubly phosphorylated dataset 2 structure and in the apo dataset 5 structure, respectively (Fig. 7E). In other structures (PBD1 and PBD2 in datasets 1, 3, and 4, and PBD2 in dataset 2) the conformations co-exist (Fig. S11A–E). Thus, there seems to be no strict correlation between the IN and OUT state of Phe559 and phospho-target binding to the PBD, at least with the isolated PBD. In fact, two other apo structures of the PBD (PDB IDs 5NN1 and 1Q4O)[11] display the Phe IN conformation, contrary to ours. Nonetheless, the two conformations of Phe559 are a further indication of the flexibility of the L2 loop.

## Discussion

We confirmed that binding of PLK1 to CENP-U is triggered by initial phosphorylation of T98 and the recruitment of PLK1 to this medium-

affinity motif. Subsequent phosphorylation of, and docking on, the otherwise PLK1-kinase-impervious site T78 leads to assembly of a very stable complex. As T78 is not an ideal PLK1 substrate, its resilience to PLK1 phosphorylation might be overcome when PLK1 is brought in close proximity. T98, on the other hand, is efficiently phosphorylated by CDK1. In a previous study, we proposed dimerization of PLK1 as a putative basis for robust binding to CENP-U that might justify its acting as a master docking site[16]. Here, we present crystal structures that confirm that the PBD can dimerize on CENP-U, but also show SEC and SEC-MALS experiments that did not detect strong incorporation of two PLK1^FL subunits on the doubly phosphorylated CENP-U. We previously reported formation of a PLK1^FL dimer on doubly-phosphorylated CENP-U in experiments of analytical ultracentrifugation (AUC)[16]. We suspect the different outcome of these previous experiments with the SEC-MALS experiments reported here may be due to sample dilution in the latter (favouring dissociation) but not in the former.

Our observations do not discount dimerization, but do not provide further support for its functional importance, as we find PLK1 binds very tightly to a single, dominant motif containing pT78 and dimerization does not increase binding affinity to CENP-U. The affinity measured for the pT78-containing Motif 1 is at least 100-fold tighter than the strongest affinities measured so far for phosphorylated motifs in vitro, including the pT98-cointaining motif. We identify Y68, in addition to F71, as a binding determinant of Motif 1 whose mutation decreases binding affinity for the PBD (Table 2). In addition, a crystal structure comprising a longer N-terminal segment of CENP-U (Dataset 2) indicated additional interactions of CENP-U with the YKK pocket of the PBD, and showed a binding mode closely reminiscent of that of the fruit fly PLK1 inhibitor MAP205 (ref. 28). Furthermore, AlphaFold (AF) modelling also suggested that in addition to this core of interactions, additional interactions may be possible at sites that relate to the pincer and cryptic pockets through the pseudo-2-fold symmetry of the PBD. Motifs in BUB1, BUBR1, and PRC1 are predicted by AF to share similar PBD binding modes to those of MAP205 and CENP-U. The AF prediction for CENP-U for the N-terminal region preceding F71 shows a different register than the one observed in the crystal structure obtained with Dataset 2, either because of inherent ambiguities in AF predictions, or because of crystal packing effects around CENP-U[37–47], which may alter the conformation of the preceding region. Be that as it may, future work will have to combine further predictions and experimental structure determination to identify crucial binding determinants of master motifs.

With slow dissociation rates, master sites may cause persistent local activation of PLK1 necessary for the phosphorylation of other targets, while providing a hierarchical organization allowing control at only a few focal points. We do not expect master sites to be very numerous; however, secondary sites generated locally may be much more numerous. Indeed, we have demonstrated that CENP-U provides the only high-affinity binding site for PLK1 in a reconstituted complex containing the 16-subunit CCAN complex and most of the 10-subunit

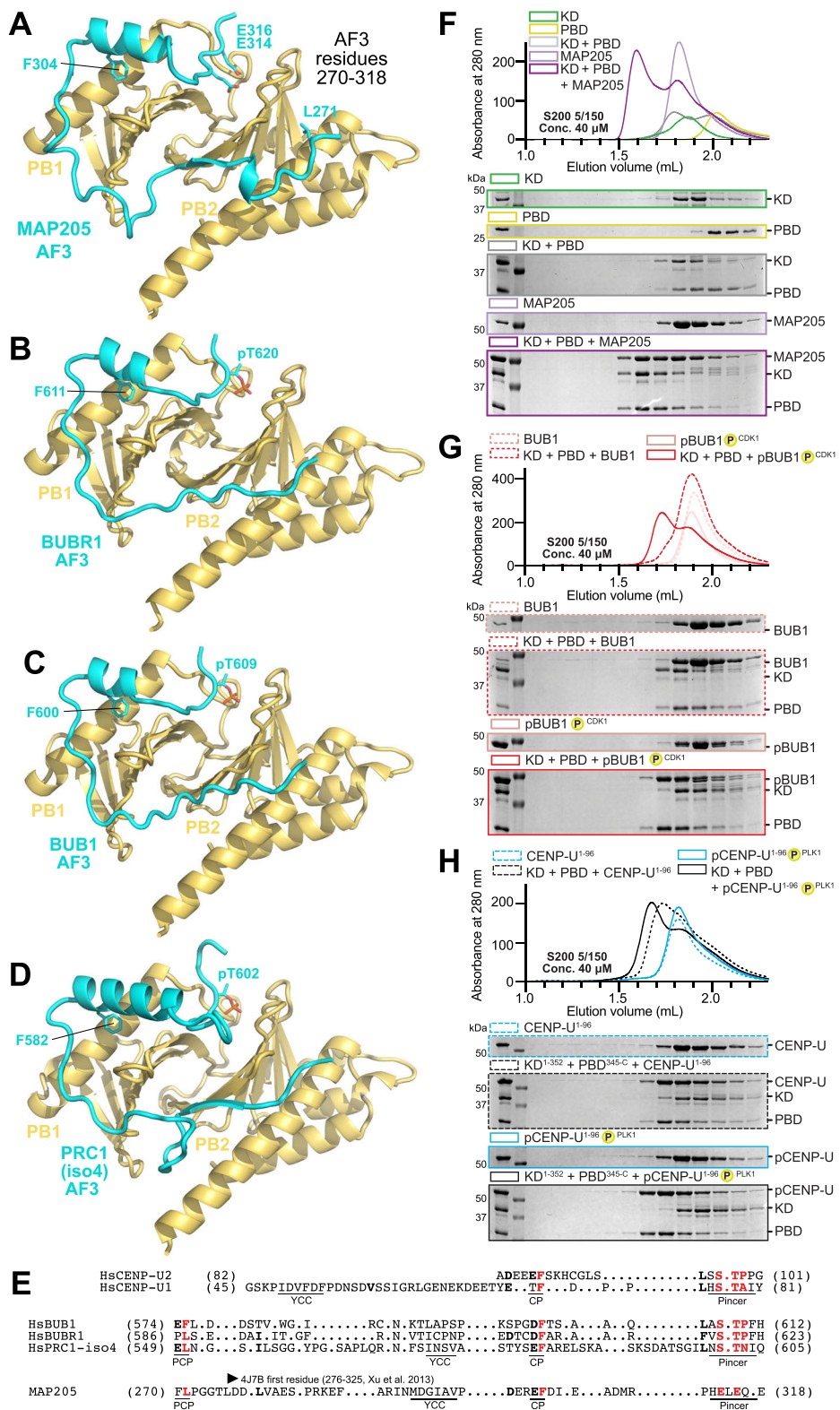

outer kinetochore complex and phosphorylated in vitro with various mitotic kinases[16].

We and others have recently shown that PLK1 localization to the kinetochore in early G1 phase depends entirely on the localization there of M18BP1[60,61]. Specifically, after anaphase onset and inactivation of CDK activity, PLK1 becomes recruited to T78 and S93 of M18BP1 through a self-primed mechanism required for deposition of the centromere-specific histone variant CENP-A[60,61]. Because CENP-U and the rest of the CCAN reside at kinetochores throughout the cell cycle, including the early G1 phase, PLK1 would be recruited even in the absence of M18BP1 if CDK1 priming weren't required for CENP-U localization. Instead, ablation of T78 and S93 phosphorylation on M18BP1 clears PLK1 from anaphase kinetochores. Thus, our current understanding of the mechanisms of PLK1 kinetochore recruitment is consistent with the claim that binding to pT78 primed by PLK1 is only possible after PLK1 docking at the neighbouring pT98 generated by

**Fig. 5 | Prediction of PLK1 master docking motifs. A** AF prediction of the PBD:MAP205 complex comprising residues not present in the crystal structure shown in Fig. 4A. L271 occupies the pseudo-cryptic pocket. **B** AF3 prediction of a segment of BUBR1 bound to the PBD, shown in the same orientation as the model in panel A, as in successive panels. **C** AF3 prediction of a segment of BUB1 bound to the PBD. **D** AF3 prediction of a segment of PRC1 (isoform 4) bound to the PBD. **E** ChimeraX[104] structure-based alignment of the indicated sequences based on the available AF and experimental models. In the underlying annotation: PCP Pseudo cryptic pocket; CP Cryptic pocket; YKK = YKK pocket; Pincer Phosphoresidue binding pocket. The start residues of MAP205 in the PDB 4J7B file is also indicated. The interpretation of the CENP-U alignment is based on the structures discussed here (not on AF models). The interpretation of BUB1, BUBR1, and PRC2 structures is based on AF models. The interpretation of MAP205 is based on structural work (4J7B) and AF modelling. **F**–**H** In (**E**), size-exclusion chromatography experiments on the indicated column. The kinase domain (KD, green) and Polo-box domain (PBD, yellow) form a complex when mixed that elutes earlier than the individual proteins (grey). MAP205 (light violet) stabilizes a stoichiometric complex of KD and PBD (dark violet). In panel F, addition of BUB1 phosphorylated by CDK1, which binds PBD, disrupts the KD:PBD complex, as shown by release of the KD. In (G), CENP-U phosphorylated with PLK1 also causes the disruption of the interaction of the PBD with the KD. Source data are provided as a Source Data file.

CDK1, a mechanism we define as relay-priming. Why CENP-U utilizes a mechanism of relay priming to generate a master docking site for PLK1, while BUB1 and BUBR1 only utilize direct CDK1 phosphorylation, remains unclear. We surmise that it might have to do with the kinetics of dephosphorylation at anaphase, but this will require further analyses. M18BP1 represents another clear example of a master docking site, because PLK1, after its recruitment there, can phosphorylate and bind additional proteins involved in the CENP-A deposition pathway. The M18BP1 master site may represent a distinct class with two adjacent phosphorylation sites that bind to a single PBD[60]. Future work will have to address differences and commonalities between these master docking motifs.

Our results have important implications for how binding of phosphopeptide ligands activates PLK1. Our work identifies the L2 loop as potential sensor of PBD engagement and as an effector displacing the kinase domain through conformational change (Fig. 6). In structures of the isolated PBD, the conformation of the L2 loop appears to be partially independent of peptide binding or of its state of phosphorylation (Fig. 7D). In all cases examined, even in absence of bound peptide, the observed conformation of the L2 loop is that expected for the open, active form where the PBD and kinase domain do not interact. On the other hand, full-length PLK1, with an unoccupied PBD, is predicted to adopt the closed conformation, independently of activation loop phosphorylation or ATP binding. We expect apo PBD to have a more flexible L2 loop, and that bound peptides of increasing affinity stabilize the L2 loop in a progressively more rigid open conformation. We speculate that part of the binding energy from PBD engagement is then harnessed to trigger the switch to an open conformation, and that the latter is retained until dissociation of the bound peptide. This mechanism, an allosteric competition between peptide and kinase domain binding to the PBD, is consistent with the apparent reduction of binding affinity of full-length PLK1 for phosphopeptides relative to the isolated PBD (Table 2). Thus, our observations are consistent with the idea that the L2 loop of PLK1 in the closed, inactive form of PLK1 adopts a higher energy conformation relative to that observed in the structures of isolated PBD domains (with or without bound ligands), which are proxies for the open form of PLK1. A strained conformation of the L2 loop in the closed form may be stabilized and compensated by reciprocal interactions with the IDL and kinase domain. When combined with the binding of a phosphopeptide, L2 repositioning may provide the energy required to dislodge the IDL and kinase domain, leading to kinase activation.

MAP205 binds to the PBD in a similar way to CENP-U and other phosphopeptides, but the outcome is that it stabilizes the closed conformation. The crucial difference in MAP205 is the replacement of the Ser-pSer/pThr-X motif with a Glu-Leu-Glu motif. The side chain of the leucine residue in this tripeptide replaces a leucine positioned at the base of the L2 loop (L490 in human PLK1 and L481 in Zebrafish), displacing the L2 loop. We speculate that displacement of the L2 loop through this mechanism increases the energy required to trigger PLK1 opening, thus stabilizing closure. L490, on the other hand, does not appear to change its position when PLK1 transitions from the closed to the open state as a consequence of phosphopeptide binding (Fig. 6C,

E). The divergence in the position of L2 residues in the open and closed states begins at the neighbouring residue L491, and becomes progressively amplified in the direction of the apex of the loop.

In conclusion, we suspect that master docking sites hold the key to robust activation of PLK1, possibly because by binding to a much larger interface, they elicit a more cooperative and durable stabilization of the open conformation. These considerations will serve as a basis for the design of new studies probing the PLK1 activation mechanism.

## Methods

### Plasmids

Plasmids encoding 6xHis-MBP-TEV-PLK1[345-603] ([MBP]PBD), 6xHis-MBP-TEV-CENP-U[58-114] ([MBP]CENP-U[58-114]), and 6xHis-TEV-PLK1 (PLK1[FL]) for expression in insect cells were available from a previous study and were used precisely as described[16]. Full-length 6xHis-TEV-PLK1 (PLK1[FL]) and CDK1:Cyclin-B:CKS1 (1:1:2) (CCC) were prepared as described in detailed protocols[16,88]. Plasmids encoding 6xHis-MBP-TEV-CENP-U[84-114] ([MBP]CENP-U[84-114]), 6xHis-MBP-TEV-CENP-U[58-83] ([MBP]CENP-U[58-83]), and 6xHis-MBP-TEV-CENP-U[39-114] ([MBP]CENP-U[39-114]) for bacterial cell expression were constructed by sub-cloning the respective PCR-amplified sequence of codon-optimized CENP-U cDNA in a modified pETDuet-1 (Novagen) in frame with coding sequences for 6xHis and PreScission-substrate–MBP–TEV-substrate. All CENP-U mutants were generated by site-directed mutagenesis directly on these templates.

### Protein expression and purification

[MBP]CENP-U constructs, [MBP]PBD, His-PLK1 1-352 K82R, MBP-CENP-U[1-96]-His, MBP-Map205[264-322]-His and MBP-BUB1[569-616]-His were expressed in *Escherichia coli* BL21 (DE3) cells. Following transformation with plasmids, the bacteria were cultured in Terrific Broth at 37 °C. Protein expression was induced using 0.2 mM IPTG ([MBP]CENP-U constructs, [MBP]PBD) or 0.1 mM IPTG (His-PLK1 1-352 K82R, MBP-CENP-U-His MBP-Map205 and MBP-BUB1) when the optical density reached approximately 0.6 and cells were incubated at 20 °C ([MBP]CENP-U constructs, [MBP]PBD) or 18 °C (His-PLK1 1-352 K82R, MBP-CENP-U-His MBP-Map205 and MBP-BUB1) for approximately 18 hours. Cells expressing PLK1[FL] and [MBP]PBD were pelleted and resuspended in Lysis buffer (50 mM HEPES pH 7.5, 500 mM NaCl, 5% (v/v) Glycerol, 10 mM Imidazole, and 1 mM TCEP; or 20 mM Bis-Tris/HCl pH 6.5 in place of 50 mM HEPES pH 7.5 in case of [MBP]PBD). After supplementation with DNase I (Roche) at 10 µg/mL and either 1 mM PMSF or 160 µg/mL HP plus protease inhibitor mix (Serva), the cell lysates were prepared by fluidization and cleared by centrifugation at 4 °C for 45 minutes. The soluble lysate was filtered through a 0.8 µm membrane and applied to a pre-equilibrated, 2× 5-ml HisTALON Cartridge containing pre-packed TALON Superflow Resin (Clontech). The column was washed with at least 10 volumes of the corresponding lysis buffer, and bound proteins were eluted using the lysis buffer supplemented with 300 mM imidazole. Eluted fractions were pooled, concentrated, and subjected to separation on a Superdex 200 16/60 size exclusion chromatography (SEC) column, pre-equilibrated in SEC buffer 1 (50 mM HEPES pH 7.5, 300 mM NaCl, 5 % (v/v) glycerol, and 1 mM TCEP) or SEC buffer 2 (20 mM Bis-tris/HCl pH

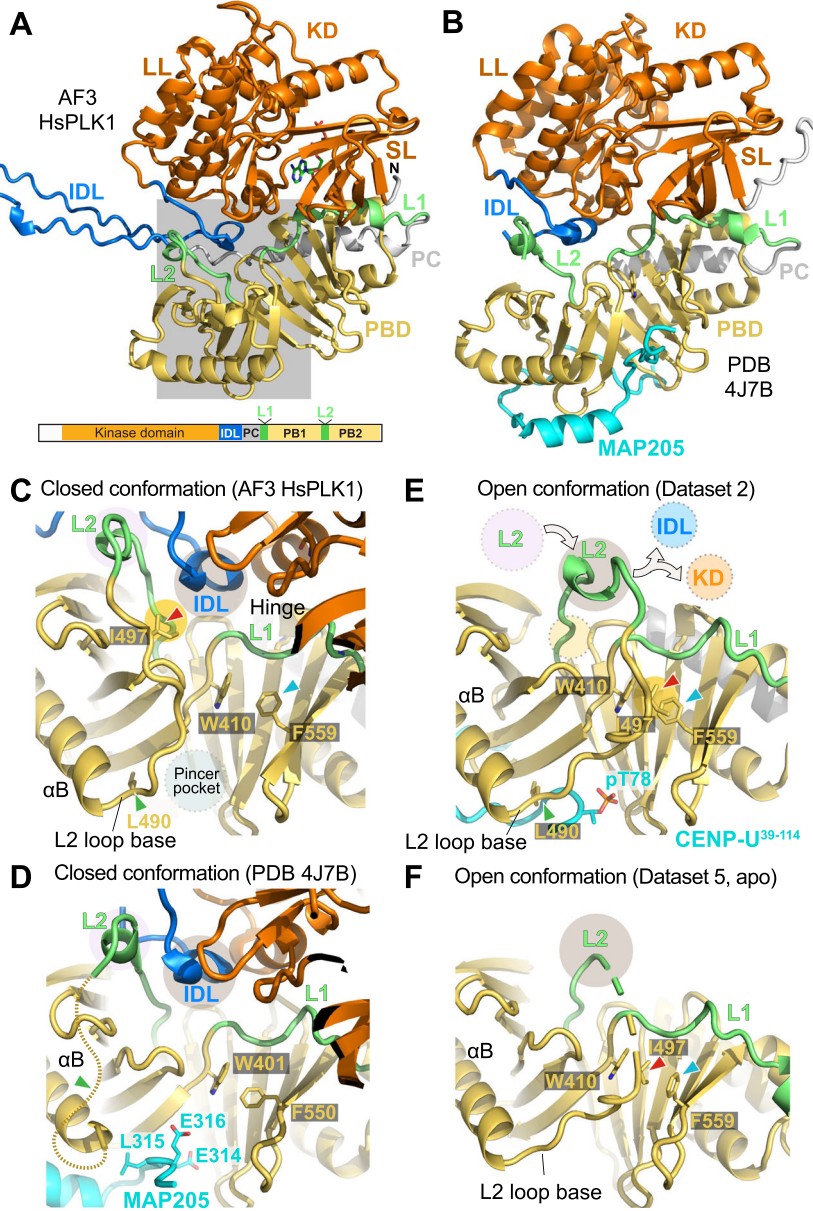

**Fig. 6 | Open and closed PLK1 conformations. A** AF modelling of the entire HsPLK1 sequence (with ATP and T210 phosphorylation) shows an arrangement that is essentially indistinguishable from that of the PLK1:MAP205 complex (4J7B). The main structural elements already introduced in Fig. 1 are repeated here for reference. The grey area is the focus of closeups shown in panels C-F. SL = Small lobe; LL = Large lobe. **B** Cartoon model of the PLK1:MAP205 complex (PDB ID 4J7B). **C** Closeup of the AF model of full-length HsPLK1, in the grey area of panel A, with rotation to highlight the main elements. The position of the L2 and IDL is indicated with pink and grey circles, respectively. The position of the base of the L2 loop next to the pincer is shown. **D** PLK1:MAP205 complex (4J7B). The dashed line represents

the expected path of the invisible base of the L2 loop. All elements are in the same position as in the predicted structure of full-length PLK1. **E** In the structure calculated with dataset 2 (as well as datasets 1, 3, and 4), the L2 loop blocks the position occupied by the IDL in the models displayed in panels C and D. In the context of full-length PLK1, the IDL would have to be displaced from its position to permit this conformation of the L2 loop. **F** Apo PBD structure calculated from dataset 5. The base of the L2 loop adopts a conformation similar to that observed in the liganded structures exemplified by dataset 2 in panel E. It follows that this conformation is not triggered by peptide binding.

6.5, 200 mM NaCl, 5 % (v/v) glycerol and 1 mM TCEP). Following inspection via 12.5% SDS-PAGE, chosen fractions were concentrated, aliquoted, flash frozen in liquid nitrogen, and stored at -80 °C. For purification of untagged PBD, a similar procedure was followed, except that the tag was removed after elution from the Talon column with His-MBP-TEV protease (produced in-house) and dialyzed for 12 hours against SEC buffer 2. PLK1 Kinase domain 1-352 K82R was expressed as a TEV cleavable 8xHis fusion, and the tag was retained throughout the purification. Bacteria from a 2-liter culture were pelleted and resuspended in lysis buffer (50 mM Hepes pH 7.5, 300 mM NaCl, 5% (v/v) glycerol, 1 mM TCEP, 2 mM PMSF). Every subsequent step was

performed in lysis buffer devoid of PMSF. The resulting soluble lysate was subjected to Cobalt-based chromatography by loading it on a 5-mL HisTALON Cartridge (Cytiva), pre-equilibrated in affinity buffer (50 mM Hepes pH 7.5, 300 mM NaCl, 5% vol/vol glycerol, 1 mM TCEP). After extensive washes, protein was eluted with 300 mM imidazole in the same buffer. The eluted sample was concentrated using an ultra-filtration centrifugal protein concentrator (MerckMillipore). The protein was finally resolved on a HiLoad 16/600 Superdex 200 pg sizing column (Cytiva), pre-equilibrated in affinity buffer, and selected fractions were concentrated, aliquoted, flash frozen in liquid nitrogen, and stored at −80 °C.

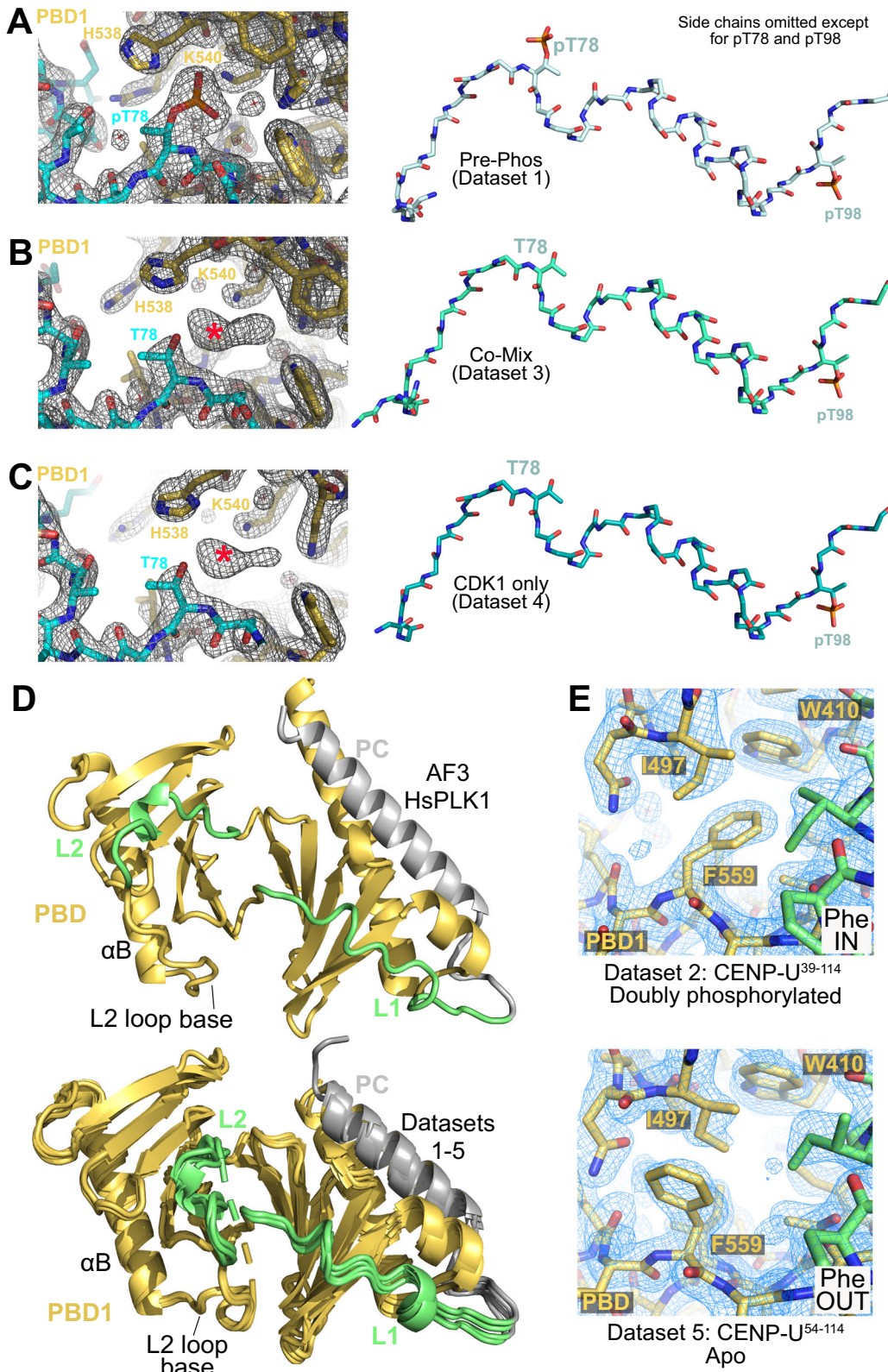

**Fig. 7 | Implications for activation mechanism of PLK1. A–C** Comparison of weighted $2F_o\text{-}F_c$ electron densities maps and models calculated with Datasets 1, 3 and 4 (displayed in (**A**, **B**, and **C**), respectively) demonstrate that despite the absence of phosphate on T78 for datasets 3 and 4, the overall structure of the bound CENP-U peptide (where side chains were omitted to improve clarity) is essentially indistinguishable from that of the doubly phosphorylated peptide. The position of a putative phosphate ion detected by anomalous X-ray scattering (AXRS) is indicated by a red asterisk. **D** Cartoon model emphasizing the different positions of the L2 loop in the AF model of full-length PLK1 (*top*; other structural elements were removed to improve clarity) and in superposed models calculated from datasets 1 to 5 (*bottom*). **E** Phe IN and Phe OUT conformation of F559 documented against the electron density in the indicated datasets. The side chain of Ile497 retains its position in the Phe IN or Phe OUT conformations of Phe559.

For 6xHis-MBP-TEV-CENP-U variants, bacterial cells were pelleted and resuspended in a Lysis buffer (50 mM HEPES pH 7.5, 500 mM NaCl, 5% (v/v) Glycerol, 10 mM Imidazole, and 1 mM TCEP). Following lysis, cell lysates were initially purified using a HisTALON Cartridge with the His-elution buffer (20 mM Bis-tris/HCl pH 6.5, 200 mM NaCl, 5% (v/v) Glycerol, 200 mM Imidazole, and 1 mM TCEP). Eluates were diluted in 9 volumes of dilution buffer (20 mM Bis-tris/HCl pH 6.5, 5% (v/v) glycerol, and 1 mM TCEP), filtered (0.2 μm membrane), and loaded onto a pre-equilibrated Resource Q anion exchange chromatography column (GE Healthcare) in 20 mM Bis-tris/HCl pH 6.5, 20 mM NaCl, 5% (v/v) glycerol, 1 mM TCEP. Proteins were then eluted with a linear gradient of 20-1000 mM NaCl, and fractions were monitored via 12.5% SDS-PAGE. Selected fractions were combined, concentrated, and subjected to further separation using a Superdex 75 16/60 SEC column (GE Healthcare) pre-equilibrated in SEC buffer 2. Following evaluation by SDS-PAGE, chosen fractions were concentrated, divided into aliquots, rapidly frozen in liquid nitrogen, and stored at -80 °C.

For purification of MBP-CENP-U-His, MBP-BUB1-His, and MBP-MAP205-His, a recently described protocol was applied[89]. The tag was retained throughout purification. Briefly, bacterial cells from 1-L cultures were collected by centrifugation and resuspended in lysis buffer (50 mM HEPES pH 7.5, 250 mM NaCl, 5% v/v glycerol, 1 mM TCEP), supplemented with HP Plus protease inhibitor mix (Serva) and DNase I (Roche). The resulting soluble lysate was subjected to Cobalt-based chromatography by loading it on a 5-mL HisTALON Cartridge (Cytiva), pre-equilibrated in lysis buffer, and after washing, the protein was eluted with 300 mM imidazole in lysis buffer. Eluted protein was pooled, concentrated, and subjected to separation on a Superdex 200 16/60 size exclusion chromatography (SEC) column, pre-equilibrated in SEC buffer (50 mM HEPES pH 7.5, 250 mM NaCl, 5% (v/v) Glycerol, and 1 mM TCEP). Selected fractions were concentrated, aliquoted, flash frozen in liquid nitrogen, and stored at −80 °C.

## In-situ phosphorylation of CENP-U constructs

To functionalize CENP-U variants in their interactions with ^MBP PBD or PLK1^FL, we employed (1) a co-mixing protocol where substrate CENP-U variants were incubated with PLK1 constructs immediately before addition of designated active kinase(s) for concurrent phosphorylation and potential complex formation overnight; or (2) a pre-phosphorylation protocol where substrate CENP-U variants were phosphorylated for 15 hours with the designated active kinase(s), and PLK1 constructs were subsequently added to allow potential complex formation. If not otherwise stated, ^MBP PBD or PLK1^FL were added in a molar ratio of 2:1 to the CENP-U variant. All steps were conducted at 4 °C.

## Analytical size exclusion chromatography

SEC analysis was conducted on a Superdex 200 5/150 column. Sample preparation, column equilibration, and elution were performed using SEC buffer (20 mM Bis-tris/HCl, pH 6.5, 250 mM NaCl, 5% (v/v) glycerol, and 1 mM TCEP). In case of samples containing PLK1^FL, the SEC buffer was modified by replacing 20 mM Bis-tris/HCl, pH 6.5, with 20 mM HEPES, pH 7.5. For the experiments in Fig. 5F−H, a SEC buffer containing 50 mM HEPES pH 7.5, 150 mM NaCl, 5% (v/v) glycerol, and 1 mM TCEP was used. Before loading on the column, a fraction of the sample was held as input for SDS-PAGE with Pro-Q™ Diamond and Coomassie blue staining. For direct interaction analysis between binding-functionalized CENP-U variants and PLK1^FL, sample inputs were verified further by Western blotting using antibodies specific to the phosphorylated Thr78^CENP-U (anti−PBIP-1(phosphor-T78) rabbit polyclonal; Abcam, Cambridge, UK; 1:1000 diluted for use) and phosphorylated Thr210^PLK1 (anti-PLK1-phospho-T210, mouse monoclonal; Biolegend #629801, 1:1000 diluted for use). All samples were eluted under isocratic conditions at 4 °C with a flow rate of 0.15 ml/min. Protein elution was monitored by UV detection at a wavelength of 280 nm, and 100 μl fractions were collected that were subsequently analysed by SDS-PAGE with TCE (2,2,2-Trichlorethanol) or Coomassie blue staining in case of samples containing PLK1^FL. Stained gels were digitalized using a BioRAD chemiDoc MP Imaging System (BioRAD).

## Multi angle light scattering

SEC-MALS analyses were performed using a 1260 Infinity II SEC system coupled with a Dawn Heleos-II System and an Optilab T-rEX Refractive Index (RI) detector (Wyatt). A Superdex 200 10/300 column (GE Healthcare) was used for the SEC elution and pre-equilibrated with Glycerol-free SEC buffer containing 20 mM HEPES pH 7.5, 300 mM NaCl and 1 mM TCEP. The column was cooled to 12 °C using a Shimadzu column cooler. Samples were prepared with a total protein concentration higher than 2 mg/mL, and of which 60 μL were loaded for the analysis. The theoretical dn/dc for each sample was calculated using SedFit. The analysis was done by Astra 7.3.2.21, using BSA for peak normalization. The molar mass for each peak was calculated using the theoretical dn/dc of the proteins and the RI signal. The data were fitted with the Zimm equation.

## Preparation of phosphorylated CENP-U variants

CENP-U variants (in 100 μM) were phosphorylated by incubation with the indicated kinase overnight at 4 °C in phosphorylation buffer (with the same composition as SEC buffer 2) supplemented with MgCl$_2$ (10 mM) and ATP (2.0 mM). CCC was used at a 1/30th of the concentration of substrate. The corresponding conversion rate of this Thr98 phosphorylation method was determined by mass spectrometry. Thr78 was phosphorylated using activated 6xHis-TEV-PLK1 (PLK1^FL) activated in vitro with Aurora A/Bora complex at a molar ratio of 200:10:1 (PLK1:Aurora A:Bora) as recently described[86]. The corresponding conversion rate of this Thr78 phosphorylation method was determined by mass spectrometry. After incubation, the reaction mixture was then concentrated and purified using a Superdex 75 16/60 SEC column (GE Healthcare) pre-equilibrated in the SEC buffer. Fractions were monitored by SDS-PAGE and Pro-Q™ Diamond staining (Invitrogen, according to the manufacturer's instructions), those containing the phosphorylated protein were concentrated, flash-frozen in liquid nitrogen, and stored at −80 °C.

CENP-U^1-96 was phosphorylated by preparing a 150 μM stock in 50 mM HEPES pH 7.5, 250 mM NaCl, 5 % (v/v) Glycerol, 1 mM TCEP, 2.5 mM ATP and 5 mM MgCl2, incubated with 750 nM 6xHis-TEV-PLK1 (PLK1^FL), 750 nM His-Aurora A, and 750 nM MBP-Bora 1-224-His for 15 hours at 10 °C. Subsequently, the phosphorylated CENP-U was separated from the other components by injection into a Superdex 200 5/150 column, concentrated, and aliquoted for binding experiments.

BUB1 was phosphorylated as recently described[89]. Briefly, 10 mg of pure protein (diluted at a 500 μM concentration) were incubated with 2.5 μM CDK1/CyclinB1 in 50 mM HEPES pH 7.5, 250 mM NaCl, 5 % (v/v) Glycerol, 1 mM TCEP, 2.5 mM ATP, and 5 mM MgCl2, for 15 hours at 10 °C. Phosphorylated BUB1 was loaded on a pre-equilibrated Superdex 10/300 200 pg sizing column (Cytiva), concentrated, aliquoted, and snap-frozen for subsequent use in binding assays.

## Preparation of CENP-U:PBD complexes for crystallographic analysis

In preparation for crystallization, we first pre-phosphorylated the relevant ^MBP CENP-U fragments on ~15 mg scale (at 40 μM reaction concentration) and incubated them with CDK1 and PLK1 for 15 hours. We then incubated the phosphorylated CENP-U with 2 equivalents of ^MBP PBD for ~3 hours in assembly buffer (20 mM Bis-tris/HCl, pH 6.5, 300 mM NaCl, 5% (v/v) glycerol, 1 mM TCEP), further concentrated the mix, and separated it on a Superdex 200 Increase 10/300 SEC column (Cytiva) pre-equilibrated in the assembly buffer. Fractions containing assembled complexes were collected and concentrated to roughly the

previous volume of the assembly mixture. The 6xHis-MBP-tagged TEV Protease (made in-house) was added at 1/20 dilution to substrates and cleavage was allowed to run for 15 hours at 4 °C. The mixture was diluted with dilution buffer (20 mM Bis-tris/HCl pH 6.5, 5% (v/v) glycerol, 1 mM TCEP) to achieve a NaCl concentration of 50 mM and loaded onto a tandem 5-mL HisTALON Cartridge (Clontech) and Resource Q anion exchange chromatography column (GE Healthcare) pre-equilibrated in 20 mM Bis-tris/HCl pH 6.5, 50 mM NaCl, 5% (v/v) glycerol, 1 mM TCEP. The complex was eluted using a linear gradient ranging from 50 to 1000 mM NaCl. Following SDS-PAGE analysis with Pro-Q™ Diamond and Coomassie blue staining, fractions containing PBD and CENP-U (the latter only visible upon phosphoprotein staining) were pooled and concentrated for a final SEC purification and buffer exchange on a Superdex 75 10/300 SEC column (GE Healthcare) pre-equilibrated in Xtal-SEC buffer (10 mM Bis-tris/HCl pH 6.5, 100 mM NaCl, 1 mM TCEP). After elution, the fractions were examined again by SDS-PAGE with Pro-Q™ Diamond phosphoprotein staining and Coomassie blue staining. The purified complex was concentrated to ~8 mg/mL and used freshly for crystallization screen. The same protocol was adapted for the co-mixing procedure.

## Crystallization and structure determination
Protein complexes used for crystallization were freshly purified. Crystallization was conducted using the sitting-drop vapour diffusion method in TTP-IQ format 96-well plates (SPT LabTech) at 20 °C. The screening-plate setup was performed utilizing a mosquito HTS robot (SPT Labtech) at 20 °C, wherein protein sample was mixed with reservoir crystallization solution in a volume ratio of 1:1 or 2:1. Crystals of the doubly phosphorylated complex containing CENP-U[58-114] (dataset 1, 9FJH) appeared within 24 hrs and kept growing for 4 days in a solution containing 0.1 M CHES pH 9.5, 0.2 M NaCl, 10% PEG8000. The same solution containing 15% (v/v) PEG400 was then used as cryoprotectant for a brief soak-in of crystals before mounting onto a cryoloop and promptly flash freezing in liquid nitrogen. Crystals of the monophosphorylated complexes (datasets 3 and 4, 9FJG and 9FGI) grew rapidly (less than 1 day) in 0.08 M NaH2(PO4), pH 6.2, 0.02 M Na3-citrate, and 18% PEG2000. 10% (v/v) PEG400 was used as cryoprotectant. Crystals of doubly phosphorylated complex containing CENP-U[39-114] (dataset 2, 9FJJ) appeared within 6 hrs and grew for 2 days in a solution containing 0.2 M tri-Lithium citrate, 20% PEG3350. 10% (v/v) glycerol was used as a cryoprotectant. Crystals of untagged PBD (apo PBD, dataset 5, 9QMO) started to form in 3 days at 4 °C in a solution of 0.2 M Na2H(PO4) and 20% PEG 3350. This reservoir solution was directly used as a cryoprotectant. Collection of native synchrotron X-ray diffraction data for an anomalous X-ray scattering assay (AXRS) for phosphorus atom determination[90] were conducted at beamline PXII X10SA at the Swiss Light Source in Villigen (Switzerland) or at the ID23-2 microfocus beamline at European Synchrotron Radiation Facility (ESRF) in Grenoble (France). Diffraction intensities were processed using XDS and scaled with XSCALE[91]. Molecular replacement (MR) using the reported crystal structure 1Q4K (PDB ID) as the search template[11] was done with MR Phaser[92], followed by map refinement with Phenix.refine[93]. Manual model building was performed with Coot[94]. The structure of the doubly phosphorylated complex was subsequently utilized for phasing all other acquired datasets. All crystallographic statistics are summarized in Table 1. X-ray structures were analysed, and corresponding presentation images were created with Pymol 2.60 software (Schrödinger LLC, New York, NY).

## Biotin labelling and Bio-layer Interferometry (BLI)
Biotin-Streptavidin (SA) chemistry was used for immobilization of proteins onto the tip surface of an OctetRed 384 sensor from Sartorius. A DNA sequence encoding a PLPETG peptide was inserted in frame at the 3' end of the coding region for chosen immobilized ligand proteins

(Table 2), and the same protocols were used for purification. The C-terminal peptide was conjugated with Gly-Gly-Gly-Gly-Biotin (Thermo Fisher Scientific) for Biotin labelling using calcium-independent 7+ sortase A[95]. Reaction was conducted for 15 hours at 4 °C in a molar loading ratio of 1:10:100 (sortase:protein:label) in the SEC buffer of the specific ligand protein. The biotinylated ligand protein was separated from the catalytic enzyme and the excess label in the mixture by SEC in SEC buffer 2 and subsequently flash-frozen in liquid nitrogen and stored at −80 °C. To minimize buffer mismatch, all proteins, including the ligand protein and analyte proteins for titration, were subjected to buffer exchange through dialysis in BLI dialysis buffer (20 mM HEPES pH 7.5, 2.5 % (v/v) Glycerol, 300 mM NaCl, 0.1 mM TCEP). Concentrations of the dialyzed proteins were re-measured immediately before performing the BLI assay. The initial highest concentration of either the ligand or any analyte for BLI assay was achieved by diluting the dialyzed protein with BLI dialysis buffer and 5x BLI additive buffer (20 mM HEPES pH 7.5, 2.5 % (v/v) Glycerol, 300 mM NaCl, 0.1 mM TCEP, 1.5 g/L BSA, 0.25% (v/v) Tween20), ensuring a final buffer containing 0.3 g/L BSA and 0.05% (v/v) Tween20. Other dilutions were performed using the BLI working buffer (20 mM HEPES pH 7.5, 2.5 % (v/v) Glycerol, 300 mM NaCl, 0.1 mM TCEP, 0.3 g/L BSA, 0.05% (v/v) Tween20) in 2- or 3-fold serial dilutions. BLI data were recorded with the Octet BLI Discover 13.0 software. The temperature was maintained at 25 °C throughout the experiments, and plates/wells were agitated at 1000 r.p.m. when sensors were immersed. The protein was immobilized on Streptavidin SA sensors (Sartorius). The data were analysed using Octet Analysis 13.0. Kinetic signals were initially fitted using a local method. In general, measurements with a local fitting $R^2$ value above 0.95 and an RSS value below 0.01 were deemed of high quality and selected for subsequent global fitting. The final affinity constant ($K_D$) was calculated as the ratio of $k_{off}$ and $k_{on}$ values given by the global fitting.

## Isothermal titration calorimetry
ITC experiments were performed with a PEAQ-ITC calorimeter (Malvern Panalytical). Protein samples were thoroughly dialyzed or subjected to gel filtration at 4 °C against the ITC buffer, containing Hepes (pH 7.5, 20 mM), NaCl (300 mM), TCEP (1 mM) and Glycerol (1%). Following concentration adjustment, the sample cell was loaded with the specified phosphorylated CENP-U construct and titrated by MBPPBD in the injection syringe at 25 °C. The stirrer speed was set to 750 rpm. As a control, MBPPBD at similar concentrations was titrated into the ITC buffer. The data was analyzed using Microcal PEAQ-ITC 1.41 (Malvern Panalytical). Control measurements were subtracted from the experimental data using line fitting of the heat peaks. The dissociation constant was determined using Microcal PEAQ-ITC 1.41 employing the one set of sites binding model[96].

## Mass spectrometry (MS)
The phosphorylation status of the CENP-U variant resulting from different in-vitro phosphorylation treatments with kinase CDK1 and/or kinase PLK1 (please refer to Method In-situ phosphorylation of CENP-U constructs above) was assessed using liquid chromatography coupled mass spectrometry (LC-MS). For the sample preparation, the incubation concentration of CENP-U is 5 µM, and that of PBD is 10 µM. The number of sample types is 7 (as shown in Fig. 1D, E), with 5 replicates per sample type ($n = 5$). The total number of samples subjected to MS analysis is 35. Among the 7 sample types, one type served as the control, in which no kinase was added during incubation prior to the digestion and purification for MS analysis. Prepared samples (incubation completed) were treated with 1 mM Dithiothreitol and alkylated with 5.5 mM chloroacetamide, followed by sequential digestions using LysC and Trypsin. After termination with TFA (0.15% (v/v)) the resulting cut peptides were purified and enriched using StageTips as previously described[97]. For MS analysis, 200 ng (CENP-U based) of the obtained

cut-peptide mixtures, after passage through a desalting cartridge with 0.1% formic acid (aq), were separated on a Pepmap C18 nanoHPLC column (U3000 nanoHPLC system, Thermo Fisher Scientific) by performing a gradient elution using 5–30% acetonitrile with 0.1% formic acid at a flow rate of 300 nl/min. The eluted liquid was directly sprayed into an Orbitrap-type mass spectrometer via a nano-electrospray source in a Q Exactive (Thermo Fisher Scientific). A data-dependent mode was applied for one survey scan, followed by up to 10 MS/MS scans[98]. To identify phospho-sites, the resulting raw files were processed with MaxQuant (version 2.2.0.0) searching for CENP-U sequences with acetylation (N-term), oxidation (M) and phosphorylation (STY) as variable modifications and carbamidomethylation (C) as fixed modification[99]. A false discovery rate cut-off of 1% was applied at the peptide and protein levels, as well as on the phosphorylation site table[99]. Relative quantification of the phosphorylated peptides of interest, such as DEETYETFDPPLHSp**T**AIYADEEEFSK for Thr78[CENP-U] and HCGLSLSSp**T**PPGK for Thr98[CENP-U], and of the unphosphorylated counterparts was performed using Skyline[100,101]. The msms.txt table of the MaxQuant search was used to build up the library, and all peak quantifications were checked manually in Skyline. If necessary, peak areas were re-integrated manually. Peaks with the highest localization probability of phosphorylation of positions Thr78 or Thr98, respectively, were extracted. MS intensities of the peptide with Thr78 phosphorylated and of the unphosphorylated counterpart were used for the analysis of PLK1-mediated phosphorylation, and so were those of Thr98 used for the analysis of CDK1-mediated phosphorylation. The phosphorylation conversion rates were determined based on the Thr-unphosphorylated peptide in samples using the formula, $1 - \frac{I_K}{I_C}$, where $I_K$ is the MS intensity of the unphosphorylated peptide from a kinase-treated sample, while $I_C$ is that from the control sample.

## Immunofluorescence assay

HeLa cell lines were maintained in Dulbecco's Modified Eagle's Medium (DMEM) supplemented with 10% tetracycline-free FBS, 50 μg/ml Penicillin/Streptomycin, and 2 mM L-glutamine (all reagents/media were from PAN Biotech). DNA fragments encoding mNeonGreen-PLK1 were sub-cloned into a pCDNA5/FRT/TO-EGFP-IRES plasmid.

HeLa cells were seeded in a 24-well plate (Sarstedt) on acid-washed, non-treated coverslips. After 18 hours, cells were transfected with 0.025-0.050 ng/μl pCDNA5 carrying a mNeonGreen-tagged PLK1 WT or mutant and Lipofectamine 2000 in a 1:50 ratio in Optimem. After a 5-minute incubation of Lipofectamine 2000 in Optimem, DNA and the reagent were mixed 1:1 and incubated for 20 minutes at 20 °C. After 24 hours, cells were directly fixed for 10 minutes with PFA 4% in PHEM buffer. After rinsing and permeabilizing with 0.5% Triton-X in PHEM for 5 min, blocking with 10% FBS, 0.1% Triton-X in PHEM was performed at room temperature for 30 minutes. Then, cells were incubated for two hours at room temperature with an anti-CENP-A primary antibody (GeneTex, mouse) diluted at 1:500 in PHEM buffer. Subsequently, coverslips were incubated with a goat anti-mouse secondary antibody conjugated to Rhodamine Red diluted at 1:200 (identifier: 115-295-003, from Jackson Immuno Research) and the DNA staining DAPI diluted at 1:5000 (Serva), for 1 hour at room temperature, in the dark. Washing steps (three times, five minutes incubation) were performed after primary antibody incubation and secondary antibody incubation with PHEM and with PHEM supplemented with 0.1% Triton-X-100 (PHEM-T) buffer, respectively. Mowiol (Calbiochem) was used as mounting media after rinsing the coverslip with ddH2O.

## Imaging and quantification

Cells were imaged on a DeltaVision Elite (GE Healthcare) deconvolution microscope, equipped with an IX71 inverted microscope (Olympus, Japan), a PLAPON x60/1.42NA oil objective (Olympus) and a pco.edge sCMOS camera (PCO-ECH Inc., USA). For each field of view, a variable number of z-sections (typically 50–70) comprising the whole DAPI signal (405 nm channel), spaced at 0.2 μm each, were taken. Fluorescence detected in the 488 nm channel was used to assess transgene expression and PLK1 kinetochore localization (mNeonGreen-PLK1), while the 555 nm channel was employed to visualize CENP-A signal. Images were converted into maximal intensity projections in SoftWoRx (Cytiva) for analysis. For each experimental repeat, approximately 100 mitotic cells were visualized. Cells with non-detectable expression levels or saturating expression levels (at a 50 ms exposure in the 488 nm channel) were discarded during image acquisition, and at least 30–50 mitotic cells (ranging from late prophase to early metaphase) were included in the subsequent analysis.

For quantification of expression levels, regions of interest (ROIs) were drawn manually on ImageJ/Fiji[102], to precisely include the whole expression signal of single cells. A background ROI was taken in parallel for each image. Integrated density, Area and Mean Gray Value of the cell ROI and background ROI were measured using Fiji., and expression levels shown in the final quantification were calculated as follows: Expression Level = Integrated density (ROI cell) - [Area (ROI cell) * MeanGrayValue (ROI background)]. To normalize the integrated density of each cell for the ROI area, the result of the previous formula was divided by the area of the cell, which was converted to a ratio by using an average cell area of 250. After obtaining comparable expression level values, a maximum and minimum threshold of 2,500,000 and 200,000, respectively, were used to further increase homogeneity of the expression levels between conditions and have the best signal-to-noise ratio. For each experimental repeat, at least 20 cells per condition were included in the final quantification.

For quantification of the PLK1 kinetochore signal, selected cells were cropped with an ImageJ/Fiji MACRO, then the signal of PLK1 overlapping with the signal of the reference channel (CENP-A signal, thresholded using Otsu's method to generate kinetochore ROIs) was measured using an automated MACRO in ImageJ/Fiji. Measurements were exported in Excel (Microsoft) and plotted with GraphPad Prism 10 (GraphPad Software). Statistical analysis was performed using the Kruskal-Wallis test (non-parametric ANOVA test) option available on GraphPad Prism 10, showing only the comparison to the control (WT) condition. To convert P-values into an asterisk-based significance system, we used the default GraphPad Prism convention: not significant (ns) = $P > 0.05$; * = $P \leq 0.05$ but $> 0.01$; ** = $P \leq 0.01$ but $> 0.001$; *** = $P \leq 0.001$ but $> 0.0001$; and **** = $P \leq 0.0001$.

Final quantification is shown in a dot plot format, with yellow dots representing the mean PLK1 kinetochore signal of single cells. The horizontal black bar indicates the median of the three biological replicates, while black dots indicate the median of each replicate. The vertical bar represents the 95% confidence interval of the three biological replicates. Figures were assembled using Adobe Illustrator 2025.

## AlphaFold modelling

Alphafold multimer 2.3.1 (AF2)[103] and Alphafold 3.0.0 (AF3)[85] were used for all structure predictions. Post-translational modifications (i.e., phosphothreonine and phosphoserine) and ligands (ATP) were defined using CCD codes TPO, SEP, and ATP. For AF2 predictions, ten models per run were ranked according to the TM score and analyzed, for AF3, five models. PAE plots were generated using a heavily modified version of Alphafold-analysis (https://github.com/grandrea/Alphafold-analysis) and used together with the pLDDT scores to assess the quality of the models. UNIPROT identifiers: P53550 PLK1_HUMAN, O43683 BUB1_HUMAN, O60566 BUB1B_HUMAN (BubR1), Q71F23 CENPU_HUMAN, O43663-4 PRC1_HUMAN Isoform 4, P23226 MA205_DROME.

## Reporting summary

Further information on research design is available in the Nature Portfolio Reporting Summary linked to this article.

## Data availability

All mass spectrometry data have been submitted to the MassIVE repository (https://massive.ucsd.edu) with ID MSV000101084 [https://doi.org/10.25345/C5QJ78C0F] and ProteomeXchange repository (https://www.proteomexchange.org) with ID PXD075442. All cell biology imaging data have been deposited in the BioImage Archive database (https://www.ebi.ac.uk/bioimage-archive) and assigned the identifier S-BIAD3056. Previously published atomic coordinate files used in this study are 4J7B (MAP205-PLK1 complex; https://doi.org/10.2210/pdb4J7B/pdb); 1Q4K (used as the search template for Molecular Replacement; [https://doi.org/10.2210/pdb1Q4K/pdb]); 5NN1 (apo PBD; [https://doi.org/10.2210/pdb5NN1/pdb]); and 1Q4O (apo PBD; [https://doi.org/10.2210/pdb1Q4O/pdb]). Newly generated atomic coordinates are 9FJH (CENP-U58-114/78pT/98pT:PBD complex, dataset 1; [https://doi.org/10.2210/pdb9FJH/pdb]); 9FJJ (CENP-U39-114/78pT/98pT:PBD complex, dataset 2; [https://doi.org/10.2210/pdb9FJJ/pdb]); 9FJG (CENP-U58-114/98pT:PBD complex, dataset 3; [https://doi.org/10.2210/pdb9FJG/pdb]); 9FJI (CENP-U58-114/98pT:PBD complex, dataset 4; [https://doi.org/10.2210/pdb9FJI/pdb]); 9QMO (PBD, dataset 5; [https://doi.org/10.2210/pdb9qmo/pdb]). Source data are provided with this paper.

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

## Acknowledgements

We would like to thank the technical staff of the Swiss Light Source (SLS) at the Paul Scherrer Institute, Villigen, Switzerland, for their support. We acknowledge the European Synchrotron Radiation Facility (ESRF) for the provision of synchrotron radiation facilities under proposal number mx2491, and we would like to thank A. Gautam for assistance and support in using beamline ID23-2.

## Author contributions

Conceptualization L.R., A.E.V., I.V., A.M. Formal Analysis L.R., I.V. Funding acquisition A.M. Investigation A.E.V., C.K., F.M., I.V., L.R., R.G., M.P., P.J., P.G., S.W. Project Administration L.R., A.M. Resources A.E.V., C.K., S.W. Supervision A.M., L.R. Visualization A.M., L.R., I.V. Writing original draft A.M., L.R. Writing - review & editing, all authors

## Funding

A.M. acknowledges funding from the Max Planck Society, the European Research Council (ERC) Synergy Grant 951430 (BIOMECANET), the Deutsche Forschungsgemeinschaft (DFG, German Research Foundation, Project-ID 424228829 – SFB1430, and the CANTAR network under the Netzwerke-NRW program. Open Access funding enabled and organized by Projekt DEAL.

## Competing interests

The authors declare no competing interests.
