## [Transparent Peer Review file · Nature Communications]

Molecular anatomy of PLK1 master docking motifs

Corresponding Author: Professor Andrea Musacchio

Version 0:

Reviewer comments:

Reviewer #1

(Remarks to the Author)

In this manuscript the interaction of the mitotic kinase Plk1 with its substrates is analysed through a combination of structural and biochemical approaches. Plk1 is a master regulator of mitosis, and this requires that Plk1 is targeted precisely to specific proteins. Targeting of Plk1 requires its polo-box domain (PBD) that binds to phosphorylated Ser/Thr residues in proteins but how specificity is obtained in the sea of mitotic phosphorylation is not fully understood. The authors initiated this current work based on their previous work identifying the Cenp-U protein as a "master" receptor for Plk1 in the CCAN complex. The authors were interested in understanding how Cenp-U recruits Plk1 and whether dimerization acted cooperatively as they had proposed. This was since two phosphorylation sites in Cenp-U, T78 and T98, bind the PBD and that Plk1 is a dimer.

Here the authors perform biochemical reconstitution of relevant Cenp-U-PBD complexes either with unphosphorylated, single or double phosphorylated Cenp-U fragments. Complexes are analysed by SEC, BMI and ITC to obtain relevant parameters of interaction and finally the structure of PBD bound to double phosphorylated Cenp-U is solved. Based on this the authors identify T78 as a high affinity binding site while T98 is a less strong binding site, and they also establish that there is no cooperative binding. However, T98 is phosphorylated by Cdk1 to prime Plk1 recruitment and Plk1 then phosphorylates T78 to generate the high affinity binding site.

From the structures the authors also identify an additional contact between Cenp-U and the PBD in that Phe residues in the proximity to the phosphorylation site engage a cryptic pocket (CP)(described in previous literature but not extensively studied). The authors also expand their work to AF models of other known PBD sequences to expand this concept of Phe residues engaging the cryptic pocket and identify a pseudo cryptic pocket (PCP) that also binds Phe.

Collectively the work shows the complexity of Plk1 interactions with substrates through additional contacts made to the PBD. Overall, the data is of high quality and the authors are honest on their conclusions and limitations of their conclusions. I am somewhat lukewarm on how much the work adds to our understanding of how Plk1 regulates mitosis as the cellular consequences of their work is not explored. I think showing that the interactions to PCP and CP are biological relevant is required for several proteins they investigate is needed for this to be an interesting study for NCOMMS. Furthermore, exploring whether the sequential recruitment mechanism described for Cenp-U is of biological relevance should be explored. If the motif 1 is engineered to higher affinity does this bypass the requirement for T78 binding site?

It is unclear to me if the AF3 models with full length Plk1 is affected by T-loop phosphorylation. Did the authors model with and without T210 phosphorylation?

The work could be expanded to the PBDs from Plk2 and Plk3 to determine if some of the principles uncovered here is specific to Plk1.

Reviewer #2

(Remarks to the Author)

Plk1 is an important, conserved mitotic Ser/Thr kinase which regulates key steps in the cell division process. Plk1 has an N-terminal kinase domain and a C-terminal Polo-box-domain (PBD). Plk1 binds to key substrates by docking of its PBD to pre-phosphorylated Ser or Thr residues in the sequence S-pS/T-X on target proteins. Both Cdk1 as well Plk1 itself have been identified as priming kinases which provide the phosphorylated docking residue. Although there is structural information available on the PBD, it is still not completely clear how Plk1 binds to its docking sites. In particular, recent work by the same lab suggested that two PLk1 molecules may dimerise and bind simultaneously to adjacent phospho-sites on a given target protein, suggesting cooperativity in PBD docking site binding. Whether this mode of binding constitutes a general mechanism for Plk1 recruitment to target sites is, however, unclear.

Using a combination of structural and biophysical methods, Ren and colleagues now further investigate the binding of Plk1 to its docking partners. They focus particularly on the CENP-U protein, which, together with Bub1, has been shown to be one of the key Plk1 binding partners on the kinetochore. CENP-U is phosphorylated at two sites, T98 by CDK1 and T78 by Plk1 itself. While the authors confirm that the PBD can bind as a dimer to CENP-U, careful biochemical analysis of the binding affinities of purified PBD to only one of the two phosphorylated sites revealed a very high binding affinity of the PBD to pT78 without the need for dimerization. This leads the authors to conclude that, in contrast to their previous suggestion, dimerization may not be an important factor in PBD binding to key docking partners. Instead, the authors present evidence of an extended interaction surface of the dominant phosphorylated docking motif with the PBD. AlphaFold modelling of PBD interactions with further known PLK1 docking partners, including Bub1, BubR1, MAP205 and PRC1 identified similar binding modes to these proteins, suggestive of a general binding mechanism.

The experiments in the manuscript are all very carefully conducted and generally convincing. Although the advance in knowledge is relatively small, the precise mechanism by which Plk1 binds to its target proteins on the kinetochore and other cellular structures, as elucidated here, is of great interest to many cell biologists and hence worth publishing.

My comments relate to the text and the citations. There are a few mis-citations or omissions:

Introduction, first paragraph: "A crucial mitotic kinase, Polo-like kinase 1 (PLK1), has emerged for its essential functions in a number of cell division events (Pintard and Archambault, 2018; Zitouni et al., 2014). For instance, PLK1 has been implicated in the regulation of spindle assembly, centrosome function, nuclear envelope breakdown, sister chromatid cohesion, kinetochore-microtubule interactions, spindle assembly checkpoint signalling, centromere propagation, and cytokinesis, among others (Nigg, 2001; Pintard and Archambault, 2018).

"Polo-like kinases and the orchestration of cell division", Barr et al., 2004, *Nat Rev Mol Cell Biol* and "Polo on the rise – from mitotic entry to cytokinesis with Plk1" Petronczki et al., 2008, *Dev Cell*, are more comprehensive and appropriate reviews of Plk1 function through the cell cycle than the ones that are currently cited and should be cited instead or in addition of the current citations.

Introduction, third paragraph: "When X is not Pro, other kinases may be involved in the generation of PBD docking sites. "Self-priming" may occur when the phosphorylating kinase is PLK1 itself (Kang et al., 2011; Kang et al., 2006; Lee et al., 2008)."

Neef et al., 2007, *Nat. Cell Biol.* should be included in this list of citations.

Introduction, 5th paragraph: "Work on the kinetochore, a large protein assembly on chromosomes that promotes microtubule binding and biorientation (Cheeseman and Desai, 2008; Musacchio and Desai, 2017), identified several kinetochore proteins as potential binding sites for the PLK1 PBD (Amin et al., 2014; Bel Borja et al., 2024; Geraghty et al., 2021; Goto et al., 2006; Kang et al., 2011; Kang et al., 2006; Kim et al., 2015; Kim et al., 2014; Lee et al., 2015; Maia et al., 2012; Matsumura et al., 2007; Nishino et al., 2006; Pouwels et al., 2014; Qi et al., 2006; Sun et al., 2012; Taylor et al., 2023; Yeh et al., 2013; Zhuo et al., 2015).

Elowe et al., 2007, *Genes Dev.* should be included in this list of citations.

Introduction, 5th paragraph: "BUB1 also recruits BUBR1 (Overlack et al., 2015; Zhang et al., 2015), which itself interacts with PLK1 after CDK-mediated phosphorylation of T620.

Elowe et al., 2007, *Genes Dev.* should be cited here.

A more general comment: I am unsure whether the term "master docking motif" is the most appropriate term to describe the hierarchical docking of Plk1 to binding partners. Would "most upstream" or similar be better?

Reviewer #3

(Remarks to the Author)

In the manuscript entitled "Molecular anatomy of PLK1 master docking motifs", the authors combine extensively biochemical and structural analysis to characterize interactions between the master docking motif of CENP-U and the PBD of PLK1, and reveal that the N-terminal extension of the canonical docking motif contributes to the high binding affinity of the master docking motif with the PBD. The present work provides a new insight into understanding the mechanism underlying the recruitment of PLK1 to the substrates and subcellular localization during the mitotic stage. On the other hand, several points should be addressed to improve the quality of the manuscript before the potential acceptance.

1) As shown in the previous study (Xu et al., 2013), MAP205 stabilizes interactions between the PBD and kinase domain of PLK1, and inhibits the activity of PLK1. Given the similar binding mode between MAP205 and CENP-U on the PBD, it should be insightful for the manuscript to characterize whether and how unphosphorylated and phosphorylated CENP-U58-114 affects the activity of PLK1.

2) As depicted in Table 2, the mutations F87A/T98V but not the wildtype sequence compromises the binding CENP-U58-114/pT78 with the PBD, is there any explanation for this result?

3) Also in Table 2, the full length PLK1 binds to CENP-U58-114/pT78 with a weaker binding affinity than that to the PBD, this effect is more prominent for CENP-U58-114/pT98 compared with CENP-U58-114/pT78 (200 fold vs 10 fold), could the solved structures or models by AF provide a rationale to this data?

4) The description of dimerization of the PBDs on CENP-U is somewhat misleading because there is no direct contact between the two PBDs on CENP-U.

5) The manuscript claims that the affinity measured for the pT78 site is at least 100-fold tighter than the strongest affinities measured so far for phosphorylated motifs in vitro, so the direct comparison between the pT78 motif of CENP-U with the reported strongest phosphorylated motifs would be preferred to support this conclusion.

6) A summarized schematic representation for the relaying priming process is preferred in the revised manuscript.

Reviewer #4

(Remarks to the Author)

This study, titled "Molecular anatomy of PLK1 master docking motifs," offers a comprehensive analysis of how the mitotic kinase Polo-like kinase 1 (PLK1) is recruited and activated at kinetochores via high-affinity interactions with specific docking motifs. Focusing primarily on the kinetochore protein CENP-U, the authors use a combination of biochemical assays, crystallography, binding kinetics, and AlphaFold3-based modeling to dissect the molecular features that define a "master docking site." They demonstrate that PLK1 recruitment to CENP-U is initiated by CDK1-mediated phosphorylation of T98, which primes the more critical phosphorylation at T78 by PLK1 itself—a process termed "relay priming." Importantly, the study shows that high-affinity binding to the PLK1 Polo-box domain (PBD) is driven not by dimerization, as previously hypothesized, but by an extended binding interface that engages multiple conserved PBD surface pockets—including a newly described "pseudo-cryptic" pocket. The pT78 motif in CENP-U binds PLK1 with ~100-fold greater affinity than pT98 and other known motifs, qualifying it as a master docking site. Structural data further reveal that similar extended interactions are predicted for other PLK1 partners such as BUB1, BUBR1, and PRC1, suggesting a shared mechanism despite sequence variability. These findings refine the molecular understanding of PLK1 spatial regulation and activation, shedding light on how master docking motifs function as focal points for robust and persistent PLK1 activity during mitosis. The study has significant implications for future research on mitotic control and the development of PLK1-targeted therapeutics. This study provides an impressive structural and biochemical dissection of PLK1 docking to CENP-U, several limitations temper the impact of its conclusions. I'd like the authors to address the points below:

1. The study focuses heavily on in vitro reconstitution and structural data, but it lacks functional cellular assays to validate the physiological relevance of its findings. For instance, the conclusion that pT78 in CENP-U is the sole high-affinity master site is not supported by mutational studies in live cells to test PLK1 recruitment or kinetochore function. Other studies (e.g., Singh et al., 2021; Nguyen et al., 2021; Chen et al., 2021) provided cellular phenotypic data to validate their identification of PLK1 docking sites. This work builds mechanistic detail but doesn't directly test its biological consequence. Can this be tested or addressed in the manuscript?

2. The paper convincingly shows dimerization is not required for high-affinity binding of PLK1 to CENP-U in vitro. However, it fails to reconcile these findings with earlier reports (including the authors' own prior work: Singh et al., 2021) that showed PLK1 dimer formation on doubly phosphorylated CENP-U using techniques like analytical ultracentrifugation. Could you please explain this point?

3. Although the paper discusses the L2 loop and its flexibility as a potential sensor for activation, it does not establish a direct causal link between binding at the master site and kinase activation. The presence of "open" L2 conformations in apo and ligand-bound PBDs leads to ambiguous conclusions about what triggers kinase activation. This should be addressed in the text.

4. The claim that other proteins like BUB1, BUBR1, and PRC1 share master docking features is based largely on similar predicted PBD engagement rather than biochemical evidence. This risks overgeneralization, especially given that PLK1 localization and activation is highly context- and phase-specific.

Version 1:

Reviewer comments:

Reviewer #1

(Remarks to the Author)

The authors have done a great job in revising the paper and I support publication. Would it be possible to have Fig S6E incorporated into Fig. 5 - it is really helpful to have a comparison of the sequences in connection to the structural models.

Reviewer #2

(Remarks to the Author)

The authors have addressed all my concerns. I am happy to support publication.

Reviewer #3

(Remarks to the Author)

I appreciate the additional experiments carried out by the authors to address the concerns from the referees, the revision has addressed most of my concerns and worth publication.

Reviewer #4

(Remarks to the Author)

the authors have addressed my points and answered my questions properly. I do not have further queries.

REVIEWER COMMENTS

We thank all reviewers for their constructive comments on the manuscript. We also apologize for the delay in resubmitting a suitably revised version. Following departure from the laboratory of the paper's first author, Dr. Long Ren, it took us some time to reorganize our efforts to complete the revision. We have now succeeded in preparing a substantially revised manuscript that addresses many of the main points raised during review, as detailed in the point-by-point response below.

Specifically, the discussion of PLK1 master docking sites has been refocused on the experimental work on CENP-U and its comparison with MAP205. In this context, we put more emphasis on an additional pocket on the PBD surface, which we term the YKK pocket, and which appears to play a conserved role in the recognition of hydrophobic residues that precede, at various distances, the phenylalanine that binds into the cryptic pocket. We have also included new biological validation experiments supporting the role of these pockets in peptide binding, as predicted by the structural work. A new size-exclusion chromatography assay demonstrates the effects of peptide binding on PLK1 kinase opening. Furthermore, we provide a rationale for the mechanism by which MAP205 inhibits PLK1.

More generally, we have streamlined the narrative to improve readability. Concomitantly, we have toned down some of the more speculative conclusions based on AlphaFold modelling, which had raised concerns during the first round of review. We hope that the reviewers will recognize the marked improvements to the manuscript from our revision.

Reviewer #1 (Remarks to the Author):

In this manuscript the interaction of the mitotic kinase Plk1 with its substrates is analysed through a combination of structural and biochemical approaches. Plk1 is a master regulator of mitosis, and this requires that Plk1 is targeted precisely to specific proteins. Targeting of Plk1 requires its polo-box domain (PBD) that binds to phosphorylated Ser/Thr residues in proteins but how specificity is obtained in the sea of mitotic phosphorylation is not fully understood. The authors initiated this current work based on their previous work identifying the Cenp-U protein as a “master” receptor for Plk1 in the CCAN complex. The authors were interested in understanding how Cenp-U recruits Plk1 and whether dimerization acted cooperatively as they had proposed. This was since two phosphorylation sites in Cenp-U, T78 and T98, bind the PBD and that Plk1 is a dimer.

Here the authors perform biochemical reconstitution of relevant Cenp-U-PBD complexes either with unphosphorylated, single or double phosphorylated Cenp-U fragments. Complexes are analysed by SEC, BMI and ITC to obtain relevant parameters of interaction and finally the structure of PBD bound to double phosphorylated Cenp-U is solved. Based on this the authors identify T78 as a high affinity binding site while T98 is a less strong binding site, and they also establish that there is no cooperative binding. However, T98 is phosphorylated by Cdk1 to prime Plk1 recruitment and Plk1 then phosphorylates T78 to generate the high affinity binding site.

From the structures the authors also identify an additional contact between Cenp-U and the PBD in that Phe residues in the proximity to the phosphorylation site engage a cryptic pocket (CP)(described in previous literature but not extensively studied). The authors also expand their work to AF models of other known PBD sequences to expand this concept of Phe residues engaging the cryptic pocket and identify a pseudo cryptic pocket (PCP) that also binds Phe.

Collectively the work shows the complexity of Plk1 interactions with substrates through additional contacts made to the PBD. Overall, the data is of high quality and the authors are honest on their conclusions and limitations of their conclusions. I am somewhat lukewarm on how much the work adds to our understanding of how Plk1 regulates mitosis as the cellular consequences of their work is not explored.

We thank the reviewer for these encouraging comments. It is true that the manuscript does not yet address the biological implication of “master” binding sites. We note that our work here was elicited by our previous cellular observations (Singh, Pesenti et al. 2021; Conti et al. 2024) supporting the existence of such “master” sites, something for which previous mechanistic work, focused on short PBD-binding peptides, offered no obvious explanation. In our future work we will address how the main interactions stabilizing such sites contribute to the recruitment of PLK1 to various subcellular structures. We note, however, that an extension of our work in that direction would require a very significant new investment, and we consider it outside the scope of an already data-rich manuscript that presents several novel observations and interpretations on the mechanism of PBD-peptide interactions.

I think showing that the interactions to PCP and CP are biological relevant is required for several proteins they investigate is needed for this to be an interesting study for NCOMMS.

We appreciate the importance of this point and we have now generated and tested the binding properties of several new PBD mutants predicted to affect the stability of the interaction with “master sites”. Specifically, in new panels in Figure 5E-G and Figure S8A-C we demonstrate the effects of mutating the CP and PCP on kinetochore localization of a PLK1 transgene. We have also modified and extended our description of these pockets, introducing the description of a highly conserved YKK pocket neighboring the cryptic pocket, and being delimited by one shared residue, Y421, whose mutation we show has a dramatic effect on kinetochore localization of PLK1.

Furthermore, exploring whether the sequential recruitment mechanism described for Cenp-U is of biological relevance should be explored. If the motif 1 is engineered to higher affinity does this bypass the requirement for T78 binding site?

Motif 1 has the highest known affinity measured for any known PBD ligand. For this reason, we don't see how we could meaningfully replace it to make it an even stronger site. We provide additional biochemical evidence (Figure 5E-G) that this PBD-binding motif of CENP-U may be even stronger than BUB1's, as it binds the PBD in vitro even in the non-phosphorylated form (clearly at the relatively high concentrations of the in vitro assays). We also note that the biological significance of the entire CENP-OPQUR complex remains to be fully elucidated. For this reason, we feel that suggesting an investigation of the biological significance of a PLK1 docking site on this complex would go beyond the scope of the current investigation. We note that these experiments will also require concomitant control on the binding of PLK1 to BUB1 and BUBR1 at kinetochores, i.e. they require a very sophisticated setup.

It is unclear to me if the AF3 models with full length Plk1 is affected by T-loop phosphorylation. Did the authors model with and without T210 phosphorylation?

Our original model of full length PLK1 was generated with AF2 and therefore it did not include T-loop phosphorylation. We have now generated four AF3 models, with and without phosphorylation of T210, and with and without ATP in the active site, and show that they are essentially identical (Figure S9A). Importantly, the two models are also essentially identical to the AF2 model we have reported, and therefore all conclusions reached with the previous model (now replaced with a new one in Figure 6A) hold with the new models.

The work could be expanded to the PBDs from Plk2 and Plk3 to determine if some of the principles uncovered here is specific to Plk1.

We thank the reviewer for this suggestion. We appreciate the importance of addressing the understudied PLK2 and PLK3 kinases, but also felt that this would extend beyond the limits of this already data-dense manuscript, and should be possibly be discussed in a dedicated study, or maybe a review article.

Reviewer #2 (Remarks to the Author):

Plk1 is an important, conserved mitotic Ser/Thr kinase which regulates key steps in the cell division process. Plk1 has an N-terminal kinase domain and a C-terminal Polo-box-domain (PBD). Plk1 binds to key substrates by docking of its PBD to pre-phosphorylated Ser or Thr residues in the sequence S-pS/T-X on target proteins. Both Cdk1 as well Plk1 itself have been identified as priming kinases which provide the phosphorylated docking residue. Although there is structural information available on the PBD, it is still not completely clear how Plk1 binds to its docking sites. In particular, recent work by the same lab suggested that two PLk1 molecules may dimerise and bind simultaneously to adjacent phospho-sites on a given target protein, suggesting cooperativity in PBD docking site binding. Whether this mode of binding constitutes a general mechanism for Plk1 recruitment to target sites is, however, unclear.

Using a combination of structural and biophysical methods, Ren and colleagues now further investigate the binding of Plk1 to its docking partners. They focus particularly on the CENP-U protein, which, together with Bub1, has been shown to be one of the key Plk1 binding partners on the kinetochore. CENP-U is phosphorylated at two sites, T98 by CDK1 and T78 by Plk1 itself. While the authors confirm that the PBD can bind as a dimer to CENP-U, careful biochemical analysis of the binding affinities of purified PBD to only one of the two phosphorylated sites revealed a very high binding affinity of the PBD to pT78 without the need for dimerization. This leads the authors to conclude that, in contrast to their previous suggestion, dimerization may not be an important factor in PBD binding to key docking partners. Instead, the authors present evidence of an extended interaction surface of the dominant phosphorylated docking motif with the PBD. Alphafold modelling of PBD interactions with further known PLk1 docking partners, including Bub1, BubR1, MAP205 and PRC1 identified similar binding modes to these proteins, suggestive of a general binding mechanism. The experiments in the manuscript are all very carefully conducted and generally convincing. Although the advance in knowledge is relatively small, the precise mechanism by which Plk1 binds to its target proteins on the kinetochore and other cellular structures, as elucidated here, is of great interest to many cell biologists and hence worth publishing.

We thank the reviewer for his/her support.

My comments relate to the text and the citations.

There are a few mis-citations or omissions: Introduction, first paragraph: “A crucial mitotic kinase, Polo-like kinase 1 (PLK1), has emerged for its essential functions in a number of cell division events (Pintard and Archambault, 2018; Zitouni et al., 2014). For instance, PLK1 has been implicated in the regulation of spindle assembly, centrosome function, nuclear envelope breakdown, sister chromatid cohesion, kinetochore-microtubule interactions, spindle assembly checkpoint signalling, centromere propagation, and cytokinesis, among others (Nigg, 2001; Pintard and Archambault, 2018).

“Polo-like kinases and the orchestration of cell division”, Barr et al., 2004, Nat Rev Mol Cell Biol and “Polo on the rise – from mitotic entry to cytokinesis with Plk1” Petronczki et al., 2008, Dev Cell, are more comprehensive and appropriate reviews of Plk1 function through the cell cycle than the ones that are currently cited and should be cited instead or in addition of the current citations.

Thank you for suggesting these reviews. We have included them alongside the ones we had originally cited.

Introduction, third paragraph: “When X is not Pro, other kinases may be involved in the generation of PBD docking sites. “Self-priming” may occur when the phosphorylating kinase is PLK1 itself (Kang et al., 2011; Kang et al., 2006; Lee et al., 2008).” Neef et al., 2007, Nat. Cell Biol. should be included in this list of citations.

Indeed. We apologize for the original omission and have now included Neef et al. 2007 in the list.

Introduction, 5th paragraph: “Work on the kinetochore, a large protein assembly on chromosomes that promotes microtubule binding and biorientation (Cheeseman and Desai, 2008; Musacchio and Desai, 2017), identified several kinetochore proteins as potential binding sites for the PLK1 PBD (Amin et al., 2014; Bel Borja et al., 2024; Geraghty et al., 2021; Goto et al., 2006; Kang et al., 2011; Kang et al., 2006; Kim et al., 2015; Kim et al., 2014; Lee et al., 2015; Maia et al., 2012; Matsumura et al., 2007; Nishino et al., 2006; Pouwels et al., 2014; Qi et al., 2006; Sun et al., 2012; Taylor et al., 2023; Yeh et al., 2013; Zhuo et al., 2015). Elowe et al., 2007, Genes Dev. should be included in this list of citations.

Thank you, we have included the Elowe et al. 2007 citation in this list as suggested.

Introduction, 5th paragraph: “BUB1 also recruits BUBR1 (Overlack et al., 2015; Zhang et al., 2015), which itself interacts with PLK1 after CDK-mediated phosphorylation of T620. Elowe et al., 2007, Genes Dev. should be cited here.

We have included the Elowe et al. 2007 citation in this list as suggested.

A more general comment: I am unsure whether the term “master docking motif” is the most appropriate term to describe the hierarchical docking of Plk1 to binding partners. Would “most upstream” or similar be better?

We have considered possible alternatives, but eventually concluded that they were not as effective. Therefore, we would like to stick to this terminology, but we have now added a clarification on page 4, where we write “...with few high-affinity master docking motifs (i.e. motifs most upstream in the pathway) being essential for initial targeting and persistent activation of PLK1”

Reviewer #3 (Remarks to the Author):

In the manuscript entitled “Molecular anatomy of PLK1 master docking motifs”, the authors combine extensively biochemical and structural analysis to characterize interactions between the master docking motif of CENP-U and the PBD of PLK1, and reveal that the N-terminal extension of the canonical docking motif contributes to the high binding affinity of the master docking motif with the PBD. The present work provides a new insight into understanding the mechanism underlying the recruitment of PLK1 to the substrates and subcellular localization during the mitotic stage.

We thank the reviewer for recognizing aspects of novelty in our work.

On the other hand, several points should be addressed to improve the quality of the manuscript before the potential acceptance.

1) As shown in the previous study (Xu et al., 2013), MAP205 stabilizes interactions between the PBD and kinase domain of PLK1, and inhibits the activity of PLK1. Given the similar binding mode between MAP205 and CENP-U on the PBD, it should be insightful for the manuscript to characterize whether and how unphosphorylated and phosphorylated CENP-U58-114 affects the activity of PLK1.

We appreciate the importance of the question. As a premise, release of the PLK1 kinase domain from the PBD may in principle directly activate the kinase (e.g. by releasing it from an inhibitory interaction with the PBD). However, it may also facilitate phosphorylation of neighboring substrates simply by allowing the kinase to explore space around it through a “leash” provided by the unfolded IDL. At present, we don’t know which, if any, of these two non-exclusive models is more relevant. Be that as it may, both models identify the detachment of the kinase domain from the PBD as a proxy for activation (or lack thereof). We have therefore carried out new experiments (Figure 5E-G and Figure S9A-C) to address activation through the proxy represented by this detachment. We confirm that the kinase and PBD domains of PLK1 have low basal affinity for each other. Binding of MAP205 stabilizes their interaction. Conversely, BUB1 and CENP-U, after phosphorylation respectively with CDK1 and PLK1, destabilize the interaction by binding to the PBD. At the protein concentrations of our in vitro assay, CENP-U destabilizes the kinase:PBD interaction even in the absence of phosphorylation, as already implied by the structural work discussed in Figure 6.

2) As depicted in Table 2, the mutations F87A/T98V but not the wildtype sequence compromises the binding CENP-U58-114/pT78 with the PBD, is there any explanation for this result?

We thank the reviewer for discussing this point. What we have observed in this comparison is a less than 3-fold decrease in binding affinity for the mutant. The overall affinity remains very high and we would be at risk of overinterpreting these relatively small differences in binding affinity, not least because the two constructs being compared have different lengths, so that the observed differences may be caused by somewhat different hydrodynamic properties.

3) Also in Table 2, the full length PLK1 binds to CENP-U58-114/pT78 with a weaker binding affinity than that to the PBD, this effect is more prominent for CENP-U58-114/pT98 compared with CENP-U58-114/pT78 (200 fold vs 10 fold), could the solved structures or models by AF provide a rationale to this data?

We suspect that the reason for this effect is that phosphopeptides binding to the PBD in full length PLK1 use part of their binding energy to displace the kinase domain (allosteric competition). In this context, we suspect that higher binding affinity makes it easier to open PLK1, i.e. the cost is proportionally higher for low affinity binders than for high affinity binders.

4) The description of dimerization of the PBDs on CENP-U is somewhat misleading because there is no direct contact between the two PBDs on CENP-U.

We have now tried to remove any potential source of misunderstanding. In particular, the sentence “Thus, Motif 1 binds the PBD with very high binding affinity even in the absence of dimerization” has been changed as “Thus, Motif 1 binds the PBD with very high binding affinity even in the absence of the neighboring Motif 2”. And the sentence “From this, we conclude that dimerization, while clearly happening, may not be a crucial factor in the creation of a master site for PLK1 on CENP-U” has been changed as “From this, we conclude that docking of two PBDs on CENP-U, while clearly happening, may not be a crucial factor in the creation of a PLK1 master docking site.”

5) The manuscript claims that the affinity measured for the pT78 site is at least 100-fold tighter than

the strongest affinities measured so far for phosphorylated motifs in vitro, so the direct comparison between the pT78 motif of CENP-U with the reported strongest phosphorylated motifs would be preferred to support this conclusion.

The comparison is based on published measurements of Sledz et al. 2011 (cited in our manuscript), who had measured a binding affinity of 250 nM by calorimetry with the pT78 motif in the context of a peptide (sequence FDPPLHSpTA) devoid of the N-terminal extension we have used. To our knowledge, this was the highest binding affinity for a PBD-peptide interaction ever reported before our study. We show affinities that are 2 orders of magnitude higher with the N-terminal extension. Our internal control is the single pT98 peptide (Table 2), which shows an affinity (400 nM), similar to that measured by Sledz et al. 2011 with the pT78 peptide, and with a similar binding mode where F87 occupies the cryptic box. Other PLK1 ligands have been reported to have lower binding affinities, as already discussed in our manuscript. We also note that our internal validation comes from measuring binding affinity with two orthogonal methodologies, BLI and ITC, obtaining closely related results.

6) A summarized schematic representation for the relaying priming process is preferred in the revised manuscript.

We have now modified Figure 1 to incorporate the representation of the relay priming mechanism, to which we refer also in the figure's legend.

Reviewer #4 (Remarks to the Author):

This study, titled "Molecular anatomy of PLK1 master docking motifs," offers a comprehensive analysis of how the mitotic kinase Polo-like kinase 1 (PLK1) is recruited and activated at kinetochores via high-affinity interactions with specific docking motifs. Focusing primarily on the kinetochore protein CENP-U, the authors use a combination of biochemical assays, crystallography, binding kinetics, and AlphaFold3-based modeling to dissect the molecular features that define a "master docking site." They demonstrate that PLK1 recruitment to CENP-U is initiated by CDK1-mediated phosphorylation of T98, which primes the more critical phosphorylation at T78 by PLK1 itself—a process termed "relay priming." Importantly, the study shows that high-affinity binding to the PLK1 Polo-box domain (PBD) is driven not by dimerization, as previously hypothesized, but by an extended binding interface that engages multiple conserved PBD surface pockets—including a newly described "pseudo-cryptic" pocket. The pT78 motif in CENP-U binds PLK1 with ~100-fold greater affinity than pT98 and other known motifs, qualifying it as a master docking site. Structural data further reveal that similar extended interactions are predicted for other PLK1 partners such as BUB1, BUBR1, and PRC1, suggesting a shared mechanism despite sequence variability. These findings refine the molecular understanding of PLK1 spatial regulation and activation, shedding light on how master docking motifs function as focal points for robust and persistent PLK1 activity during mitosis. The study has significant implications for future research on mitotic control and the development of PLK1-targeted therapeutics. This study provides an impressive structural and biochemical dissection of PLK1 docking to CENP-U.

We thank the reviewer for these encouraging comments on our manuscript.

Several limitations temper the impact of its conclusions. I'd like the authors to address the points below:

The reviewer also expressed several concerns that we have addressed as follows

1. The study focuses heavily on in vitro reconstitution and structural data, but it lacks functional cellular assays to validate the physiological relevance of its findings. For instance, the conclusion that pT78 in

CENP-U is the sole high-affinity master site is not supported by mutational studies in live cells to test PLK1 recruitment or kinetochore function. Other studies (e.g., Singh et al., 2021; Nguyen et al., 2021; Chen et al., 2021) provided cellular phenotypic data to validate their identification of PLK1 docking sites. This work builds mechanistic detail but doesn't directly test its biological consequence. Can this be tested or addressed in the manuscript?

As already indicated in our response to reviewer 1, we agree that the manuscript does not yet address the biological implications of “master” binding sites. We note, however, that our work here was elicited by our previous cellular observations (Singh, Pesenti et al. 2021; Conti et al. 2024) that such “master” sites appear to exist, something for which there was no biochemical explanation. Singh, Pesenti et al. 2021 addressed the question of specificity of PLK1 recruitment to the kinetochore. The main conclusion of that work is that BUB1 and CENP-U, both of which are predicted to contain master sites, are the main primary kinetochore receptors of PLK1. BUBR1 is also clearly contributing with a high affinity PLK1 receptor site, but being dependent on BUB1 for kinetochore recruitment, its localization there disappears when BUB1 is depleted. We note that experiments of biological validation on CENP-U are highly complex. To test localization, one would have to control the concomitant recruitment to the BUB1/BUBR1 axis. This would require co-depletion of BUB1, and the replacement of CENP-OPQUR, which is very stably bound to the kinetochore, with appropriate mutants. We have now built a new cell line to remove CENP-OPQUR acutely in mitosis, and used it in preliminary experiments to confirm the results in Singh, Pesenti, et al. as a premise for future work. However, we do not have the capacity to address this question at present and also feel that the manuscript is already very heavy on data.

2. The paper convincingly shows dimerization is not required for high-affinity binding of PLK1 to CENP-U in vitro. However, it fails to reconcile these findings with earlier reports (including the authors' own prior work: Singh et al., 2021) that showed PLK1 dimer formation on doubly phosphorylated CENP-U using techniques like analytical ultracentrifugation. Could you please explain this point?

We thank the reviewer for raising this point. In AUC, the concentration remains at ~1 mg/ml in 400 μ l throughout the entire run, and therefore we suspect that the 2:1 complex remains stable due to the concentration. On the contrary, in SEC-MALS experiments, we dilute 100 μ l of the original 1 mg/ml solution onto a 25 ml column. We had already included our interpretation of this difference in the original manuscript, and we have slightly refined it: “We previously reported formation of a PLK1FL dimer on doubly-phosphorylated CENP-U in experiments of analytical ultracentrifugation (AUC). We suspect the different outcome of these previous experiments with the SEC-MALS experiments reported here may be due to sample dilution (favouring dissociation) in the latter but not in the former.”

3. Although the paper discusses the L2 loop and its flexibility as a potential sensor for activation, it does not establish a direct causal link between binding at the master site and kinase activation. The presence of “open” L2 conformations in apo and ligand-bound PBDs leads to ambiguous conclusions about what triggers kinase activation. This should be addressed in the text.

We have now revised and extended this section in the Discussion. We point to the role of residues in MAP205 in the stabilization of the closed conformation. We postulate that the closed state of the L2 loop in the closed conformation is kept in place by interactions between the PBD and kinase domain, also through the IDL. We hypothesize that ligand binding provides the energy required to revert this, thus releasing the L2 loop from its high-energy state. This explains why any isolated PBD domain, whether bound to a peptide or not, has the open L2 conformation in the absence of the kinase domain. We discuss this in the text.

4. The claim that other proteins like BUB1, BUBR1, and PRC1 share master docking features is based largely on similar predicted PBD engagement rather than biochemical evidence. This risks overgeneralization, especially given that PLK1 localization and activation is highly context- and phase-specific.

We agree with the reviewer that more work is needed to investigate the significance of this model. However, our claims are based on a chain of observations, first and foremost the fact that removing BUB1 and CENP-U leads to loss of PLK1 from mitotic kinetochores (as originally demonstrated by Singh, Pesenti et al. 2021). Second, the motifs that bind BUB1, BUBR1, and PRC1, while divergent, are related. Third, they are predicted by AF to bind to an extensive interface of the PBD. We do not see other sequences bind with the same mode. The CENP-U Motif 1 binds the PBD with the highest affinities ever measured for PBD ligands, with multiple sources of evidence confirming this. It seems to us therefore that our claims are based on strong supporting evidence. However, we have now included new in vivo evidence showing that mutations at the cryptic box and at the YKK box dramatically affect kinetochore localization of PLK1 in mitosis.

REVIEWERS' COMMENTS

Reviewer #1 (Remarks to the Author):

The authors have done a great job in revising the paper and I support publication. Would it be possible to have Fig S6E incorporated into Fig. 5 - it is really helpful to have a comparison of the sequences in connection to the structural models.

We thank the reviewer for supporting publication of our revised manuscript. We have also followed the reviewer's suggestion and moved panel S6E into Figure 5 (as panel E). The legends and figure calls have been modified accordingly.

Reviewer #2 (Remarks to the Author):

The authors have addressed all my concerns. I am happy to support publication.

We thank the reviewer for supporting publication of our revised manuscript

Reviewer #3 (Remarks to the Author):

I appreciate the additional experiments carried out by the authors to address the concerns from the referees, the revision has addressed most of my concerns and worth publication.

We thank the reviewer for supporting publication of our revised manuscript

Reviewer #4 (Remarks to the Author):

The authors have addressed my points and answered my questions properly. I do not have further queries.

We thank the reviewer for supporting publication of our revised manuscript